# Privacy Amplification by Mixing and Diffusion Mechanisms

**Borja Balle**

**Gilles Barthe**
MPI for Security and Privacy
IMDEA Software Institute

**Marco Gaboardi**
Boston University

**Joseph Geumlek**
University of California, San Diego

## Abstract

A fundamental result in differential privacy states that the privacy guarantees of a mechanism are preserved by any post-processing of its output. In this paper we investigate under what conditions stochastic post-processing can amplify the privacy of a mechanism. By interpreting post-processing as the application of a Markov operator, we first give a series of amplification results in terms of uniform mixing properties of the Markov process defined by said operator. Next we provide amplification bounds in terms of coupling arguments which can be applied in cases where uniform mixing is not available. Finally, we introduce a new family of mechanisms based on diffusion processes which are closed under post-processing, and analyze their privacy via a novel heat flow argument. On the applied side, we generalize the analysis of "privacy amplification by iteration" in Noisy SGD and show it admits an exponential improvement in the strongly convex case, and study a mechanism based on the Ornstein–Uhlenbeck diffusion process which contains the Gaussian mechanism with optimal post-processing on bounded inputs as a special case.

## 1 Introduction

Differential privacy (DP) [15] has arisen in the last decade into a strong de-facto standard for privacy-preserving computation in the context of statistical analysis. The success of DP is based, at least in part, on the availability of robust building blocks (e.g., the Laplace, exponential and Gaussian mechanisms) together with relatively simple rules for analyzing complex mechanisms built out of these blocks (e.g., composition and robustness to post-processing). The inherent tension between privacy and utility in practical applications has sparked a renewed interest into the development of further rules leading to tighter privacy bounds. A trend in this direction is to find ways to measure the privacy introduced by sources of randomness that are not accounted for by standard composition rules. Generally speaking, these are referred to as *privacy amplification* rules, with prominent examples being amplification by *subsampling* [9, 18, 20, 6, 5, 8, 2, 27], *shuffling* [16, 10, 3] and *iteration* [17].

Motivated by these considerations, in this paper we initiate a systematic study of privacy amplification by *stochastic post-processing*. Specifically, given a DP mechanism $M$ producing (probabilistic) outputs in $\mathbb{X}$ and a Markov operator $K$ defining a stochastic transition between $\mathbb{X}$ and $\mathbb{Y}$, we are interested in measuring the privacy of the post-processed mechanism $K \circ M$ producing outputs in $\mathbb{Y}$. The standard post-processing property of DP states that $K \circ M$ is at least as private as $M$. Our goal is to understand under what conditions the post-processed mechanism $K \circ M$ is *strictly* more private than $M$. Roughly speaking, this amplification should be non-trivial when the operator $K$ "forgets" information about the distribution of its input $M(D)$. Our main insight is that, at least when $\mathbb{Y} = \mathbb{X}$,

the forgetfulness of $K$ from the point of view of DP can be measured using similar tools to the ones developed to analyze the speed of convergence, i.e. *mixing*, of the Markov process associated with $K$.

In this setting, we provide three types of results, each associated with a standard method used in the study of convergence for Markov processes. In the first place, Section 3 provides DP amplification results for the case where the operator $K$ satisfies a uniform mixing condition. These include standard conditions used in the analysis of Markov chains on discrete spaces, including the well-known Dobrushin coefficent and Doeblin's minorization condition [19]. Although in principle uniform mixing conditions can also be defined in more general non-discrete spaces [12], most Markov operators of interest in $\mathbb{R}^d$ do not exhibit uniform mixing since the speed of convergence depends on how far apart the initial inputs are. Convergence analyses in this case rely on more sophisticated tools, including Lyapunov functions [22], coupling methods [21] and functional inequalities [1].

Following these ideas, Section 4 investigates the use of coupling methods to quantify privacy amplification by post-processing under Rényi DP [23]. These methods apply to operators given by, e.g., Gaussian and Laplace distributions, for which uniform mixing does not hold. Results in this section are intimately related to the privacy amplification by iteration phenomenon studied in [17] and can be interpreted as extensions of their main results to more general settings. In particular, our analysis unpacks the *shifted* Rényi divergence used in the proofs from [17] and allows us to easily track the effect of iterating arbitrary noisy Lipschitz maps. As a consequence, we show an exponential improvement on the privacy amplification by iteration of Noisy SGD in the strongly convex case which follows from applying this generalized analysis to *strict* contractions.

Our last set of results concerns the case where $K$ is replaced by a family of operators $(P_t)_{t \geq 0}$ forming a *Markov semigroup* [1]. This is the natural setting for *continuous-time* Markov processes, and includes *diffusion* processes defined in terms of stochastic differential equations [25]. In Section 5 we associate (a collection of) diffusion mechanisms $(M_t)_{t \geq 0}$ to a diffusion semigroup. Interestingly, these mechanisms are, by construction, closed under post-processing in the sense that $P_s \circ M_t = M_{s+t}$. We show the Gaussian mechanism falls into this family – since Gaussian noise is closed under addition – and also present a new mechanism based on the Ornstein-Uhlenbeck process which has better mean squared error than the standard Gaussian mechanism (and matches the error of the optimally post-processed Gaussian mechanism with bounded inputs). Our main result on diffusion mechanisms provides a generic Rényi DP guarantee based on an intrinsic notion of sensitivity derived from the geometry induced by the semigroup. The proof relies on a heat flow argument reminiscent of the analysis of mixing in diffusion processes based on functional inequalities [1].

## 2  Background

We start by introducing notation and concepts that will be used throughout the paper. We write $[n] = \{1, \ldots, n\}$, $a \wedge b = \min\{a, b\}$ and $[a]_+ = \max\{a, 0\}$.

**Probability.**  Let $\mathbb{X} = (\mathbb{X}, \Sigma, \lambda)$ be a measurable space with sigma-algebra $\Sigma$ and base measure $\lambda$. We write $\mathcal{P}(\mathbb{X})$ to denote the set of probability distributions on $\mathbb{X}$. Given a probability distribution $\mu \in \mathcal{P}(\mathbb{X})$ and a measurable event $E \subseteq \mathbb{X}$ we write $\mu(E) = \mathsf{P}[X \in E]$ for a random variable $X \sim \mu$, denote its expectation under $f : \mathbb{X} \to \mathbb{R}^d$ by $\mathsf{E}[f(X)]$, and can get back its distribution as $\mu = \mathsf{Law}(X)$. Given two distributions $\mu, \nu$ (or, in general, arbitrary measures) we write $\mu \ll \nu$ to denote that $\mu$ is absolutely continuous with respect to $\nu$, in which case there exists a Radon-Nikodym derivative $\frac{d\mu}{d\nu}$. We shall reserve the notation $p_\mu = \frac{d\mu}{d\lambda}$ to denote the density of $\mu$ with respect to the base measure. We also write $\mathcal{C}(\mu, \nu)$ to denote the set of couplings between $\mu$ and $\nu$; i.e. $\pi \in \mathcal{C}(\mu, \nu)$ is a distribution on $\mathcal{P}(\mathbb{X} \times \mathbb{X})$ with marginals $\mu$ and $\nu$. The support of a distribution is $\mathrm{supp}(\mu)$.

**Markov Operators.**  We will use $\mathcal{K}(\mathbb{X}, \mathbb{Y})$ to denote the set of Markov operators $K : \mathbb{X} \to \mathcal{P}(\mathbb{Y})$ defining a stochastic transition map between $\mathbb{X}$ and $\mathbb{Y}$ and satisfying that $x \mapsto K(x)(E)$ is measurable for every measurable $E \subseteq \mathbb{Y}$. Markov operators act on distributions $\mu \in \mathcal{P}(\mathbb{X})$ on the left through $(\mu K)(E) = \int K(x)(E)\mu(dx)$, and on functions $f : \mathbb{Y} \to \mathbb{R}$ on the right through $(Kf)(x) = \int f(y)K(x, dy)$, which can also be written as $(Kf)(x) = \mathsf{E}[f(X)]$ with $X \sim K(x)$. The kernel of a Markov operator $K$ (with respect to $\lambda$) is the function $k(x, \cdot) = \frac{dK(x)}{d\lambda}$ associating with $x$ the density of $K(x)$ with respect to a fixed measure.

**Divergences.**  A popular way to measure dissimilarity between distributions is to use Csiszár divergences $\mathsf{D}_\phi(\mu \| \nu) = \int \phi(\frac{d\mu}{d\nu})d\nu$, where $\phi : \mathbb{R}_+ \to \mathbb{R}$ is convex with $\phi(1) = 0$. Taking

$\phi(u) = \frac{1}{2}|u - 1|$ yields the total variation distance $\mathsf{TV}(\mu, \nu)$, and the choice $\phi(u) = [u - e^{\varepsilon}]_+$ with $\varepsilon \geq 0$ gives the hockey-stick divergence $\mathsf{D}_{e^{\varepsilon}}$, which satisfies

$$\mathsf{D}_{e^{\varepsilon}}(\mu\|\nu) = \int \left[\frac{d\mu}{d\nu} - e^{\varepsilon}\right]_+ d\nu = \int [p_\mu - e^{\varepsilon} p_\nu]_+ d\lambda = \sup_{E \subseteq \mathbb{X}} (\mu(E) - e^{\varepsilon}\nu(E)) \ .$$

It is easy to check that $\varepsilon \mapsto \mathsf{D}_{e^{\varepsilon}}(\mu\|\nu)$ is monotonically decreasing and $\mathsf{D}_1 = \mathsf{TV}$. All Csiszár divergences satisfy joint convexity $\mathsf{D}((1 - \gamma)\mu_1 + \gamma\mu_2\|(1 - \gamma)\nu_1 + \gamma\nu_2) \leq (1 - \gamma)\mathsf{D}(\mu_1\|\nu_1) + \gamma\mathsf{D}(\mu_2\|\nu_2)$ and the data processing inequality $\mathsf{D}(\mu K\|\nu K) \leq \mathsf{D}(\mu\|\nu)$ for any Markov operator $K$. Rényi divergences[1] are another way to compare distributions. For $\alpha > 1$ the Rényi divergence of order $\alpha$ is defined as $\mathsf{R}_\alpha(\mu\|\nu) = \frac{1}{\alpha-1} \log \int (\frac{d\mu}{d\nu})^\alpha d\nu$, and also satisfies the data processing inequality. Finally, to measure similarity between $\mu, \nu \in \mathcal{P}(\mathbb{R}^d)$ we sometimes use the $\infty$-Wasserstein distance:

$$\mathsf{W}_\infty(\mu, \nu) = \inf_{\pi \in \mathcal{C}(\mu,\nu)} \inf\{w \geq 0 : \|X - Y\| \leq w \text{ holds almost surely for } (X, Y) \sim \pi\} \ .$$

**Differential Privacy.** A mechanism $M : \mathbb{D}^n \to \mathcal{P}(\mathbb{X})$ is a randomized function that takes a dataset $D \in \mathbb{D}^n$ over some universe of records $\mathbb{D}$ and returns a (sample from) distribution $M(D)$. We write $D \simeq D'$ to denote two databases differing in a single record. We say that $M$ satisfies[2] $(\varepsilon, \delta)$-DP if $\sup_{D \simeq D'} \mathsf{D}_{e^{\varepsilon}}(M(D)\|M(D')) \leq \delta$ [15]. Furthermore, we say that $M$ satisfies $(\alpha, \epsilon)$-RDP if $\sup_{D \simeq D'} \mathsf{R}_\alpha(M(D)\|M(D')) \leq \epsilon$ [23].

## 3 Amplification From Uniform Mixing

We start our analysis of privacy amplification by stochastic post-processing by considering settings where the Markov operator $K$ satisfies one of the following uniform mixing conditions.

**Definition 1.** Let $K \in \mathcal{K}(\mathbb{X}, \mathbb{Y})$ be a Markov operator, $\gamma \in [0, 1]$ and $\varepsilon \geq 0$. We say that $K$ is:

(1) $\gamma$-*Dobrushin* if $\sup_{x,x'} \mathsf{TV}(K(x), K(x')) \leq \gamma$,

(2) $(\gamma, \varepsilon)$-*Dobrushin* if $\sup_{x,x'} \mathsf{D}_{e^{\varepsilon}}(K(x)\|K(x')) \leq \gamma$,

(3) $\gamma$-*Doeblin* if there exists a distribution $\omega \in \mathcal{P}(\mathbb{Y})$ such that $K(x) \geq (1 - \gamma)\omega$ for all $x \in \mathbb{X}$,

(4) $\gamma$-*ultra-mixing* if for all $x, x' \in \mathbb{X}$ we have $K(x) \ll K(x')$ and $\frac{dK(x)}{dK(x')} \geq 1 - \gamma$.

Most of these conditions arise in the context of mixing analyses in Markov chains. In particular, the Dobrushin condition can be tracked back to [13], while Doeblin's condition was introduced earlier [14] (see also [24]). Ultra-mixing is a strengthening of Doeblin's condition used in [12]. The $(\gamma, \varepsilon)$-Dobrushin is, on the other hand, new and is designed to be a generalization of Dobrushin tailored for amplification under the hockey-stick divergence.

It is not hard to see that Dobrushin's is the weakest among these conditions, and in fact we have the implications summarized in Figure 1 (see Lemma 9). This explains why the amplification bounds in the following result are increasingly stronger, and in particular why the first two only provide amplification in $\delta$, while the last two also amplify the $\varepsilon$ parameter.

**Theorem 1.** *Let $M$ be an $(\varepsilon, \delta)$-DP mechanism. For a given Markov operator $K$, the post-processed mechanism $K \circ M$ satisfies:*

(1) $(\varepsilon, \delta')$-*DP with $\delta' = \gamma\delta$ if $K$ is $\gamma$-Dobrushin,*

(2) $(\varepsilon, \delta')$-*DP with $\delta' = \gamma\delta$ if $K$ is $(\gamma, \tilde{\varepsilon})$-Dobrushin with[3] $\tilde{\varepsilon} = \log(1 + \frac{e^{\varepsilon}-1}{\delta})$,*

(3) $(\varepsilon', \delta')$-*DP with $\varepsilon' = \log(1 + \gamma(e^{\varepsilon} - 1))$ and $\delta' = \gamma(1 - e^{\varepsilon'-\varepsilon}(1 - \delta))$ if $K$ is $\gamma$-Doeblin,*

(4) $(\varepsilon', \delta')$-*DP with $\varepsilon' = \log(1 + \gamma(e^{\varepsilon} - 1))$ and $\delta' = \gamma\delta e^{\varepsilon'-\varepsilon}$ if $K$ is $\gamma$-ultra-mixing.*

A few remarks about this result are in order. First we note that (2) is stronger than (1) since the monotonicity of hockey-stick divergences implies $\mathsf{TV} = \mathsf{D}_1 \geq \mathsf{D}_{e^{\tilde{\varepsilon}}}$. Also note how in the results above we always have $\varepsilon' \leq \varepsilon$, and in fact the form of $\varepsilon'$ is the same as obtained under amplification

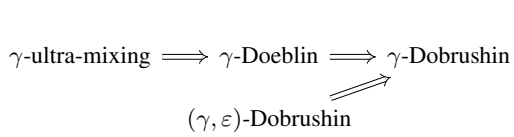

| Mixing Condition | Local DP Condition |
|---|---|
| $\gamma$-Dobrushin | $(0,\gamma)$-LDP |
| $(\gamma,\varepsilon)$-Dobrushin | $(\varepsilon,\gamma)$-LDP |
| $\gamma$-Doeblin | Blanket condition[4] |
| $\gamma$-ultra-mixing | $(\log\frac{1}{1-\gamma},0)$-LDP |

Figure 1: Implications between mixing conditions

Table 1: Relation between mixing conditions and local DP

by subsampling when, e.g., a $\gamma$-fraction of the original dataset is kept. This is not a coincidence since the proofs of (3) and (4) leverage the *overlapping mixtures* technique used to analyze amplification by subsampling in [2]. However, we note that for (3) we can have $\delta' > 0$ even with $\delta = 0$. In fact the Doeblin condition only leads to an amplification in $\delta$ if $\gamma \leq \frac{\delta e^\varepsilon}{(1-\delta)(e^\varepsilon-1)}$.

We conclude this section by noting that the conditions in Definition 1, despite being quite natural, might be too stringent for proving amplification for DP mechanisms on, say, $\mathbb{R}^d$. One way to see this is to interpret the operator $K : \mathbb{X} \to \mathcal{P}(\mathbb{Y})$ as a mechanism and to note that the uniform mixing conditions on $K$ can be rephrased in terms of *local DP* (LDP) [18] properties (see Table 1 for property[4] translations)where the supremum is taken over *any pair* of inputs (instead of neighboring ones). This motivates the results on next section, where we look for finer conditions to prove amplification by stochastic post-processing.

## 4 Amplification From Couplings

In this section we turn to coupling-based proofs of amplification by post-processing under the Rényi DP framework. Our first result is a measure-theoretic generalization of the shift-reduction lemma in [17] which does not require the underlying space to be a normed vector space. The main differences in our proof are to use explicit couplings instead of the shifted Rényi divergence which implicitly relies on the existence of a norm (through the use of $W_\infty$), and replace the identity $U + W - W = U$ between random variables which depends on the vector-space structure with a transport operators $H_\pi$ and $H_{\pi'}$ which satisfy $\mu H_{\pi'} H_\pi = \mu$ in a general measure-theoretic setting.

Given a coupling $\pi \in \mathcal{C}(\mu,\nu)$ with $\mu,\nu \in \mathcal{P}(\mathbb{X})$, we construct a *transport* Markov operator $H_\pi : \mathbb{X} \to \mathcal{P}(\mathbb{X})$ with kernel[5] $h_\pi(x,y) = \frac{p_\pi(x,y)}{p_\mu(x)}$, where $p_\pi = \frac{d\pi}{d\lambda \otimes \lambda}$ and $p_\mu = \frac{d\mu}{d\lambda}$. It is immediate to verify from the definition that $H_\pi$ is a Markov operator satisfying the transport property $\mu H_\pi = \nu$ (see Lemma 16).

**Theorem 2.** *Let $\alpha \geq 1$, $\mu,\nu \in \mathcal{P}(\mathbb{X})$ and $K \in \mathcal{K}(\mathbb{X},\mathbb{Y})$. For any distribution $\omega \in \mathcal{P}(\mathbb{X})$ and coupling $\pi \in \mathcal{C}(\omega,\mu)$ we have*

$$\mathsf{R}_\alpha(\mu K \| \nu K) \leq \mathsf{R}_\alpha(\omega \| \nu) + \sup_{x \in \mathrm{supp}(\nu)} \mathsf{R}_\alpha((H_\pi K)(x) \| K(x)) \ . \tag{1}$$

Note that this result captures the data-processing inequality for Rényi divergences since taking $\omega = \mu$ and the identity coupling yields $\mathsf{R}_\alpha(\mu K \| \nu K) \leq \mathsf{R}_\alpha(\mu \| \nu)$. The next examples illustrate the use of this theorem to obtain amplification by operators corresponding to the addition of Gaussian and Laplace noise.

**Example 1** (Iterated Gaussian)**.** We can show that (1) is tight and equivalent to the shift-reduction lemma [17] on $\mathbb{R}^d$ by considering the simple scenario of adding Gaussian noise to the output of a Gaussian mechanism. In particular, suppose $M(D) = \mathcal{N}(f(D), \sigma_1^2 I)$ for some function $f$ with global $L_2$-sensitivity $\Delta$ and the Markov operator $K$ is given by $K(x) = \mathcal{N}(x, \sigma_2^2 I)$. The post-processed mechanism is given by $(K \circ M)(D) = \mathcal{N}(f(D), (\sigma_1^2 + \sigma_2^2)I)$, which satisfies $(\alpha, \frac{\alpha\Delta^2}{2(\sigma_1^2+\sigma_2^2)})$-RDP. We now show how this result also follows from Theorem 2. Given two datasets $D \simeq D'$ we write $\mu = M(D) = \mathcal{N}(u, \sigma_1^2 I)$ and $\nu = M(D') = \mathcal{N}(v, \sigma_1^2 I)$ with $\|u - v\| \leq \Delta$. We

take $\omega = \mathcal{N}(w, \sigma_1^2 I)$ for some $w$ to be determined later, and couple $\omega$ and $\mu$ through a translation $\tau = u - w$, yielding a coupling $\pi$ with $p_\pi(x, y) \propto \exp(-\frac{\|x-w\|^2}{2\sigma_1^2})\mathbb{I}[y = x + \tau]$ and a transport operator $H_\pi$ with kernel $h_\pi(x, y) = \mathbb{I}[y = x + \tau]$. Plugging these into (1) we get

$$\mathsf{R}_\alpha(\mu K \| \nu K) \le \frac{\alpha \|w - v\|^2}{2\sigma_1^2} + \sup_{x \in \mathbb{R}^d} \mathsf{R}_\alpha(K(x + \tau) \| K(x)) = \frac{\alpha}{2}\left( \frac{\|w - v\|^2}{\sigma_1^2} + \frac{\|u - w\|^2}{\sigma_2^2} \right) \ .$$

Finally, taking $w = \theta u + (1 - \theta)v$ with $\theta = (1 + \frac{\sigma_2^2}{\sigma_1^2})^{-1}$ yields $\mathsf{R}_\alpha(\mu K \| \nu K) \le \frac{\alpha \Delta^2}{2(\sigma_1^2 + \sigma_2^2)}$.

**Example 2** (Iterated Laplace)**.** To illustrate the flexibility of this technique, we also apply it to get an amplification result for iterated Laplace noise, in which Laplace noise is added to the output of a Laplace mechanism. We begin by noting a negative result that there is no amplification in the $(\varepsilon, 0)$-DP regime.

**Lemma 3.** *Let $M(D) = \mathrm{Lap}(f(D), \lambda_1)$ for some function $f : \mathbb{D} \to \mathbb{R}$ with global $L_1$-sensitivity $\Delta$ and let the Markov operator $K$ be given by $K(x) = \mathrm{Lap}(x, \lambda_2)$. The post-processed mechanism $(K \circ M)$ does not achieve $(\varepsilon, 0)$-DP for any $\varepsilon < \frac{\Delta}{\max\{\lambda_1, \lambda_2\}}$. Note that $M$ achieves $(\frac{\Delta}{\lambda_1}, 0)$-DP and $K(f(D))$ achieves $(\frac{\Delta}{\lambda_2}, 0)$-DP.*

However, the iterated Laplace mechanism $K \circ M$ above still offers additional privacy in the relaxed RDP setting. An application of (1) allows us to identify some of this improvement. Recall from [23, Corollary 2] that $M$ satisfies $(\alpha, \frac{1}{\alpha - 1} \log g_\alpha(\frac{\Delta}{\lambda_1}))$-RDP with $g_\alpha(z) = \frac{\alpha}{2\alpha - 1} \exp(z(\alpha - 1)) + \frac{\alpha - 1}{2\alpha - 1} \exp(-z\alpha)$. As in Example 1, we take $\omega = \mathrm{Lap}(w, \lambda_1)$ for some $w$ to be determined later, and couple $\omega$ and $\mu$ through a translation $\tau = u - w$. Through (1) we obtain

$$\mathsf{R}_\alpha(\mu K \| \nu K) \le \frac{1}{\alpha - 1} \log \left( g_\alpha \left( \frac{|w - v|}{\lambda_1} \right) \right) + \sup_{x \in \mathbb{R}} \mathsf{R}_\alpha(K(x + \tau) \| K(x))$$
$$= \frac{1}{\alpha - 1} \log \left( g_\alpha \left( \frac{|w - v|}{\lambda_1} \right) g_\alpha \left( \frac{|u - w|}{\lambda_2} \right) \right) \ .$$

In the simple case where $\lambda_1 = \lambda_2$, an amplification result is observed from the log-convexity of $g_\alpha$, since $g_\alpha(a) g_\alpha(b) \le g_\alpha(a + b)$. When $\lambda_1 \ne \lambda_2$, certain values of $w$ still result in amplification, but they depend nontrivially on $\alpha$. However, we also observe that this improvement vanishes as $\alpha \to \infty$, since the necessary convexity also vanishes. In the limit, the lowest upper bound offered by (1) for $\mathsf{R}_\infty$ (which reduces to $(\varepsilon, 0)$-DP) matches the $\frac{\Delta}{\max\{\lambda_1, \lambda_2\}}$ result of Lemma 3.

**Example 3** (Lipschitz Kernel)**.** As a warm-up for the results in Section 4.1, we now re-work Example 1 with a slightly more complex Markov operator. Suppose $\psi$ is an $L$-Lipschitz map[6] and let $K(x) = \mathcal{N}(\psi(x), \sigma_2^2 I)$. Taking $M$ to be the Gaussian mechanism from Example 1, we will show that the post-processed mechanism $K \circ M$ satisfies $(\alpha, \frac{\alpha \Delta^2}{2\sigma_*^2})$-RDP with $\sigma_*^2 = \sigma_1^2 + \frac{\sigma_2^2}{L^2}$. To prove this bound, we instantiate the notation from Example 1, and use the same coupling strategy to obtain

$$\mathsf{R}_\alpha(\mu K \| \nu K) \le \frac{\alpha}{2}\left( \frac{\|w - v\|^2}{\sigma_1^2} + \sup_{x \in \mathbb{R}^d} \frac{\|\psi(x + \tau) - \psi(x)\|^2}{\sigma_2^2} \right) \le \frac{\alpha}{2}\left( \frac{\|w - v\|^2}{\sigma_1^2} + \frac{L^2\|u - w\|^2}{\sigma_2^2} \right) \ ,$$

where the second inequality uses the Lipschitz property. As before, the result follows from taking $w = \theta u + (1 - \theta)v$ with $\theta = (1 + \frac{\sigma_2^2}{L^2\sigma_1^2})^{-1}$. This example shows that we get amplification (i.e. $\sigma_*^2 > \sigma_1^2$) for any $L < \infty$ and $\sigma_2 > 0$, although the amount of amplification decreases as $L$ grows. On the other hand, for $L < 1$ the amplification is stronger than just adding Gaussian noise (Example 1).

## 4.1 Amplification by Iteration in Noisy Projected SGD with Strongly Convex Losses

Now we use Theorem 2 and the computations above to show that the proof of privacy amplification by iteration [17, Theorem 22] can be extended to explicitly track the Lipschitz coefficients in a "noisy iteration" algorithm. In particular, this allows us to show an exponential improvement on the rate of privacy amplification by iteration in noisy SGD when the loss is strongly convex. To obtain this result we first provide an *iterated* version of Theorem 2 in $\mathbb{R}^d$ with Lipschitz Gaussian

kernels. This version of the analysis introduces an explicit dependence on the $W_\infty$ distances along an "interpolating" path between the initial distributions $\mu, \nu \in \mathcal{P}(\mathbb{R}^d)$ which could later be optimized for different applications. In our view, this helps to clarify the intuition behind the previous analysis of amplification by iteration.

**Theorem 4.** *Let $\alpha \geq 1$, $\mu, \nu \in \mathcal{P}(\mathbb{R}^d)$ and let $\mathbb{K} \subseteq \mathbb{R}^d$ be a convex set. Suppose $K_1, \ldots, K_r \in \mathcal{K}(\mathbb{R}^d, \mathbb{R}^d)$ are Markov operators where $Y_i \sim K_i(x)$ is obtained as[7] $Y_i = \Pi_{\mathbb{K}}(\psi_i(x) + Z_i)$ with $Z_i \sim \mathcal{N}(0, \sigma^2 I)$, where the maps $\psi_i : \mathbb{K} \to \mathbb{R}^d$ are $L$-Lipschitz for all $i \in [r]$. For any $\mu_0, \mu_1, \ldots, \mu_r \in \mathcal{P}(\mathbb{R}^d)$ with $\mu_0 = \mu$ and $\mu_r = \nu$ we have*

$$\mathsf{R}_\alpha(\mu K_1 \cdots K_r \| \nu K_1 \cdots K_r) \leq \frac{\alpha L^2}{2\sigma^2} \sum_{i=1}^{r} L^{2(r-i)} \mathsf{W}_\infty(\mu_i, \mu_{i-1})^2 \ . \tag{2}$$

*Furthermore, if $L \leq 1$ and $\mathsf{W}_\infty(\mu, \nu) = \Delta$, then*

$$\mathsf{R}_\alpha(\mu K_1 \cdots K_r \| \nu K_1 \cdots K_r) \leq \frac{\alpha \Delta^2 L^{r+1}}{2r\sigma^2} \ . \tag{3}$$

Note how taking $L = 1$ in the bound above we obtain $\frac{\alpha \Delta^2}{2r\sigma^2} = O(1/r)$, which matches [17, Theorem 1]. On the other hand, for $L$ strictly smaller than 1, the analysis above shows that the amplification rate is $O(L^{r+1}/r)$ as a consequence of the maps $\psi_i$ being strict contractions, i.e. $\|\psi_i(x) - \psi_i(y)\| < \|x - y\|$. For $L > 1$ this result is not useful since the sum will diverge; however, the proof could easily be adapted to handle the case where each $\psi_i$ is $L_i$-Lipschitz with some $L_i > 1$ and some $L_i < 1$. We now apply this result to improve the per-person privacy guarantees of noisy projected SGD (Algorithm 1) in the case where the loss function is smooth and strongly convex.

---

**Algorithm 1:** Noisy Projected Stochastic Gradient Descent — NoisyProjSGD$(D, \ell, \eta, \sigma, \xi_0)$

**Input:** Dataset $D = (z_1, \ldots, z_n)$, loss function $\ell : \mathbb{K} \times \mathbb{D} \to \mathbb{R}$, learning rate $\eta$, noise parameter $\sigma$, initial distribution $\xi_0 \in \mathcal{P}(\mathbb{K})$

Sample $x_0 \sim \xi_0$

**for** $i \in [n]$ **do**
$\quad \lfloor \ x_i \leftarrow \Pi_{\mathbb{K}}(x_{i-1} - \eta(\nabla_x \ell(x_{i-1}, z_i) + Z))$ with $Z \sim \mathcal{N}(0, \sigma^2 I)$

**return** $x_n$

---

A function $f : \mathbb{K} \subseteq \mathbb{R}^d \to \mathbb{R}$ defined on a convex set is $\beta$-*smooth* if it is continuously differentiable and $\nabla f$ is $\beta$-Lipschitz, i.e., $\|\nabla f(x) - \nabla f(y)\| \leq \beta \|x - y\|$, and is $\rho$-strongly convex if the function $g(x) = f(x) - \frac{\rho}{2} \|x\|^2$ is convex. When we say that a loss function $\ell : \mathbb{K} \times \mathbb{D} \to \mathbb{R}$ satisfies a property (e.g. smoothness) we mean the property is satisfied by $\ell(\cdot, z)$ for all $z \in \mathbb{D}$. Furthermore, we recall from [17] that a mechanism $M : \mathbb{D}^n \to \mathbb{X}$ satisfies $(\alpha, \epsilon)$-*RDP at index $i$* if $\mathsf{R}_\alpha(M(D) \| M(D')) \leq \epsilon$ holds for any pair of databases $D$ and $D'$ *differing on the $i$th coordinate*.

**Theorem 5.** *Let $\ell : \mathbb{K} \times \mathbb{D} \to \mathbb{R}$ be a $C$-Lipschitz, $\beta$-smooth, $\rho$-strongly convex loss function. If $\eta \leq \frac{2}{\beta + \rho}$, then NoisyProjSGD$(D, \ell, \eta, \sigma, \xi_0)$ satisfies $(\alpha, \alpha\epsilon_i)$-RDP at index $i$, where $\epsilon_n = \frac{2C^2}{\sigma^2}$ and $\epsilon_i = \frac{2C^2}{(n-i)\sigma^2}(1 - \frac{2\eta\beta\rho}{\beta+\rho})^{\frac{n-i+1}{2}}$ for $1 \leq i \leq n - 1$.*

Since [17, Theorem 23] shows that for smooth Lipschitz loss functions the guarantee at index $i$ of NoisyProjSGD is given by $\epsilon_i = O(\frac{C^2}{(n-i)\sigma^2})$, our result provides an exponential improvement in the strongly convex case. This implies, for example, that using the technique in [17, Corollary 31] one can show that, in the strongly convex setting, running $\Theta(\log(d))$ additional iterations of NoisyProjSGD on *public* data is enough to attain (up to constant factors) the same optimization error as non-private SGD while providing privacy for all individuals.

## 5 Diffusion Mechanisms

Now we go beyond the analysis from previous sections and simultaneously consider a family of Markov operators $\mathbf{P} = (P_t)_{t \geq 0}$ indexed by a continuous parameter $t$ and satisfying the semigroup

property $P_t P_s = P_{t+s}$. Such **P** is called a *Markov semigroup* and can be used to define a family of output perturbation mechanisms $M_t^f(D) = P_t(f(D))$ which are closed under post-processing by **P** in the sense that $P_s \circ M_t^f = M_{t+s}^f$. The semigroup property greatly simplifies the analysis of privacy amplification by post-processing, since, for example, if we show that $M_t^f$ satisfies $(\alpha, \epsilon(t))$-RDP, then this immediately provides RDP guarantees for any post-processing of $M_t$ by any number of operators in **P**. The main result of this section provides such privacy analysis for mechanisms arising from symmetric diffusion Markov semigroups in Euclidean space. We will show this class includes the well-known Gaussian mechanism, and also identify another interesting mechanism in this class arising from the Ornstein-Uhlenbeck diffusion process.

Roughly speaking, a *diffusion* Markov semigroup $\mathbf{P} = (P_t)_{t\geq 0}$ on $\mathbb{R}^d$ corresponds to the case where $X_t \sim P_t(x)$ defines a Markov process $(X_t)_{t\geq 0}$ arising from a (time-homogeneous Itô) *stochastic differential equation* (SDE) of the form $X_0 = x$ and $dX_t = u(X_t)dt + v(X_t)dW_t$, where $W_t$ is a standard $d$-dimensional Wiener process, and the *drift* $u : \mathbb{R}^d \to \mathbb{R}^d$ and *diffusion* $v : \mathbb{R}^d \to \mathbb{R}^{d \times d}$ coefficients satisfy appropriate regularity assumptions.[8] In this paper, however, we shall follow [1] and take a more abstract approach to Markov diffusion semigroups. We synthesize this approach by making a number of hypotheses on **P** that we discuss after introducing two core concepts from the theory of Markov semigroups.

In the context of a Markov semigroup **P**, the action of the Markov operators $P_t$ on functions can be used to define the *generator* $L$ of the semigroup as the operator given by $Lf = \frac{d}{dt}(P_t f)|_{t=0}$. In particular, for a diffusion semigroup arising from the SDE $dX_t = u(X_t)dt + v(X_t)dW_t$ it is well-known that one can write the generator as $Lf = \langle u, \nabla f \rangle + \frac{1}{2}\langle vv^\top, H(f) \rangle$, where $H(f)$ is the Hessian of $f$ and the second term is a Frobenius inner product. Using the generator one also defines the so-called *carré du champ* operator $\Gamma(f, g) = \frac{1}{2}(L(fg) - fLg - gLf)$. This operator is bilinear and non-negative in the sense that $\Gamma(f) \triangleq \Gamma(f, f) \geq 0$. The *carré du champ* operator operator can be interpreted as a device to measure how far $L$ is from being a first-order differential operator, since, e.g., if $L = \sum_i a_i \frac{\partial}{\partial x_i}$ then $L(fg) = fLg + gLf$ and therefore $\Gamma(f, g) = 0$. The operator $\Gamma$ can also be related to notions of curvature/contractivity of the underlying semigroup [1]. Below we illustrate these concepts with the example of Brownian motion; but first we formally state our assumptions on the semigroup.

**Assumption 1.** Suppose the Markov semigroup $\mathbf{P} = (P_t)_{t \geq 0} \subset \mathcal{K}(\mathbb{R}^d, \mathbb{R}^d)$ satisfies the following:

(1) There exists a unique non-negative invariant measure $\lambda$; that is, $\lambda P_t = \lambda$ for all $t \geq 0$. When the invariant measure is finite we normalize it to be a probability measure.

(2) The operators $P_t$ admit a symmetric kernel $p_t(x, y) = p_t(y, x)$ with respect to the invariant measure. Equivalently, the invariant measure $\lambda$ is reversible for the Markov process $X_t$.

(3) The generator $L$ satisfies the diffusion property $L\phi(f) = \phi'(f)Lf + \phi''(f)\Gamma(f)$ for any differentiable $\phi : \mathbb{R} \to \mathbb{R}$. This is a chain rule property saying that $L$ is a second-order differential operator without constant terms.

**Example 4** (Brownian Motion)**.** The simplest diffusion process is the Brownian motion given by the simple SDE $dX_t = \sqrt{2}dW_t$., which corresponds to the semigroup **P** given by $P_t(x) = \mathcal{N}(x, 2t)$. In this case, the mechanism $M_t^f(D) = P_t(f(D))$ is a Gaussian mechanism with variance $\sigma^2 = 2t$ and therefore satisfies $(\alpha, \frac{\alpha\Delta^2}{4t})$-RDP, where $\Delta$ is the global $L_2$-sensitivity of $f$. A direct substitution with $u = 0$ and $v = \sqrt{2}I$ shows that the semigroup's generator is the standard Laplacian in $\mathbb{R}^d$, $L = \nabla^2 = \sum_{i=1}^d \frac{\partial^2}{\partial x_i^2}$, and a simple calculation yields the expression $\Gamma(f, g) = \langle \nabla f, \nabla g \rangle$ for the carré du champ operator. Now we check that **P** satisfies the conditions in Assumption 1. First, we recall that Brownian motion has the Lebesgue measure $\lambda$ on $\mathbb{R}^d$ as its unique invariant measure; this happens to be a non-finite measure. With respect to $\lambda$, the semigroup has kernel $p_t(x, y) \propto \exp(-\frac{\|x-y\|^2}{4t})$ which is clearly symmetric. Finally, we use the chain rule for the gradient to verify that

$$Lf = \nabla^2 \phi(f) = \nabla(\phi'(f)\nabla f) = \phi''(f)\langle \nabla f, \nabla f \rangle + \phi'(f)\nabla^2 f = \phi''(f)\Gamma(f) + \phi'(f)Lf \ .$$

Now we turn to the main result of this section, which provides a privacy analysis for the diffusion mechanism $M_t^f$ associated with an arbitrary symmetric diffusion Markov semigroup. The key insight

behind this result is that the carré du champ operator of the semigroup provides a measure $\Lambda(t)$ of *intrinsic sensitivity* for the mechanism $M_t^f$ defined as:

$$\Lambda(t) = \sup_{D \simeq D'} \int_t^\infty \kappa_{f(D),f(D')}(s)ds \;, \quad \text{where} \quad \kappa_{x,x'}(t) = \sup_{y \in \mathbb{R}^d} \Gamma \left( \log \frac{p_t(x,y)}{p_t(x',y)} \right) \;.$$

**Theorem 6.** *Let $f : \mathbb{D}^n \to \mathbb{R}^d$ and let $\mathbf{P} = (P_t)_{t \geq 0}$ by a Markov semigroup on $\mathbb{R}^d$ satisfying Assumption 1. If the mechanism $M_t^f(D) = P_t(f(D))$ has intrinsic sensitivity $\Lambda(t)$, then it satisfies $(\alpha, \alpha\Lambda(t))$-RDP for any $\alpha > 1$ and $t > 0$.*

**Example 5** (Brownian Motion, Continued). To illustrate the use of Theorem 6 we show how it can be used to recover the privacy guarantees of the Gaussian mechanism through its connection with Brownian motion. We let $\mathbf{P}$ be the semigroup from Example 4 and start by using $\Gamma(f) = \|\nabla f\|^2$ to compute $\kappa_{x,x'}(t)$ as follows:

$$\Gamma \left( \log \frac{p_t(x,y)}{p_t(x',y)} \right) = \left\| \nabla_y \left( \frac{\|x' - y\|^2 - \|x - y\|^2}{4t} \right) \right\|^2 = \frac{\|x - x'\|^2}{4t^2} \;.$$

Now we use $\int_t^\infty \frac{1}{s^2} ds = \frac{1}{t}$ and $\Delta^2 = \sup_{D \simeq D'} \|f(D) - f(D')\|^2$ to see that the mechanism associated with $\mathbf{P}$ has intrinsic sensitivity $\Lambda(t) = \frac{\Delta^2}{4t}$, yielding the privacy guarantee from Example 4.

## 5.1 The Ornstein-Uhlenbeck Mechanism

Beyond Brownian motion, another well-known diffusion process is the *Ornstein-Uhlenbeck* process with parameters $\theta, \rho > 0$ given by the SDE $dX_t = -\theta X_t dt + \sqrt{2}\rho dW_t$. This diffusion process is associate with the semigroup $\mathbf{P} = (P_t)_{t \geq 0}$ given by $P_t(x) = \mathcal{N}(e^{-\theta t}x, \frac{\rho^2}{\theta}(1 - e^{-2\theta t})I)$. One interpretation of this diffusion process is to think of $X_t$ as a Brownian motion with variance $\rho^2$ applied to a mean reverting flow that pulls a particle towards the origin at a rate $\theta$. In particular, the mechanism $M_t^f(D)$ is given by releasing $e^{-\theta t}f(D) + \mathcal{N}(0, \frac{\rho^2}{\theta}(1 - e^{-2\theta t}))$.

Taking the limit $t \to \infty$ one sees that the (unique) invariant measure of $\mathbf{P}$ is the Gaussian distribution $\lambda = \mathcal{N}(0, \frac{\rho^2}{\theta}I)$. From the SDE characterization of this process it is easy to check that its generator is $Lf = \rho^2 \nabla^2 f - \theta\langle x, \nabla f \rangle$ and the associated carré du champ operator is $\Gamma(f, g) = \rho^2 \langle \nabla f, \nabla g \rangle$. Thus, $\mathbf{P}$ satisfies conditions (1) and (3) in Assumption 1. To check the symmetry condition we apply a change of measure to the Gaussian density $\tilde{p}_t(x, y)$ of $P_t$ with respect to the Lebesgue measure to get its density w.r.t. $\lambda$:

$$p_t(x, y) = \frac{\tilde{p}_t(x, y)}{\tilde{p}_\lambda(y)} \propto \frac{\exp\left(-\frac{\theta\|y - e^{-\theta t}x\|^2}{2\rho^2(1 - e^{-2\theta t})}\right)}{\exp\left(-\frac{\theta\|y\|^2}{2\rho^2}\right)} = \exp\left(-\theta\frac{\|x\|^2 - 2e^{\theta t}\langle x, y \rangle + \|y\|^2}{2\rho^2(e^{2\theta t} - 1)}\right) \;,$$

where $\tilde{p}_\lambda$ is the density of $\lambda$ w.r.t. the Lebesgue measure. Thus, Theorem 6 yields the following.

**Corollary 7.** *Let $f : \mathbb{D}^n \to \mathbb{R}^d$ have global $L_2$-sensitivity $\Delta$ and $\mathbf{P} = (P_t)_{t \geq 0}$ be the Ornstein-Uhlenbeck semigroup with parameters $\theta, \rho$. For any $\alpha > 1$ and $t > 0$ the mechanism $M_t^f(D) = P_t(f(D))$ satisfies $(\alpha, \alpha\Lambda(t))$-RDP with $\Lambda(t) = \frac{\theta\Delta^2}{2\rho^2(e^{2\theta t} - 1)}$.*

The Ornstein-Uhlenbeck mechanism is not an unbiased mechanism since $\mathsf{E}[M_t^f(D)] = e^{-\theta t}f(D)$. This bias is the reason why the privacy guarantee in Corollary 7 exhibits a rate $O(e^{-2\theta t})$, while, for example, the Brownian motion mechanism only exhibits a rate $O(t^{-1})$. In particular, the Ornstein-Uhlenbeck mechanism achieves its privacy not only by introducing noise, but also by shrinking $f(D)$ towards a data-independent point (the origin in this case); this effectively corresponds to reducing the sensitivity of $f$ from $\Delta$ to $e^{-\theta t}\Delta$. This provides a way to trade-off variance and bias in the mean-squared error (MSE) incurred by privately releasing $f(D)$ in a similar way that can be achieved by post-processing the Gaussian mechanism when $f(D)$ is known to be bounded.

To formalize this result we define the *mean squared error* $\mathcal{E}_{\mathrm{OU}}(\theta, \rho, t)$ of the Ornstein-Uhlenbeck mechanism with parameters $\theta, \rho$ at time $t$, which is given by:

$$\mathcal{E}_{\mathrm{OU}}(\theta, \rho, t) \triangleq \mathsf{E}[\|f(D) - M_t^f(D)\|^2] = (1 - e^{-\theta t})^2\|f(D)\|^2 + \frac{d\rho^2}{\theta}(1 - e^{-2\theta t}) \;. \quad (4)$$

Similarly, we define $\mathcal{E}_{\mathrm{GM}}(\theta, \rho, t)$ as the mean squared error of a Gaussian mechanism with the same privacy guarantees as $M_t^f$ with parameters $\theta, \rho$. In particular, we have $\mathcal{E}_{\mathrm{GM}}(\theta, \rho, t) = d\tilde{\sigma}^2$, where $\tilde{\sigma}^2 \triangleq \frac{\rho^2(e^{2\theta t}-1)}{\theta}$ (cf. Corollary 7). We also note the post-processed Gaussian mechanism (PGM) $D \mapsto \beta(f(D) + \mathcal{N}(0, \tilde{\sigma}^2 I))$ which multiplies the output by a scalar $\beta$ optimized to minimize the MSE under the condition $\|f(D)\| \leq R$ yields error $\mathcal{E}_{\mathrm{PGM}}(\theta, \rho, t) \leq \mathcal{E}_{\mathrm{GM}}(\theta, \rho, t)(1 + \frac{d\tilde{\sigma}^2}{R^2})^{-1}$.

**Theorem 8.** *Suppose $f : \mathbb{D}^n \to \mathbb{R}^d$ has global $L_2$-sensitivity $\Delta$ and satisfies $\sup_D \|f(D)\| \leq R$. If $\theta R^2 \leq 4d\rho^2$ then we have $\frac{\mathcal{E}_{\mathrm{OU}}(\theta,\rho,t)}{\mathcal{E}_{\mathrm{GM}}(\theta,\rho,t)} \leq 1$ for all $t \geq 0$ and $\lim_{t\to\infty} \frac{\mathcal{E}_{\mathrm{OU}}(\theta,\rho,t)}{\mathcal{E}_{\mathrm{GM}}(\theta,\rho,t)} = 0$. In particular, taking $\theta = \log\left(1 + \frac{d\Delta^2}{2\epsilon R^2}\right)$ and $\rho^2 = \frac{\theta\Delta^2}{2\epsilon(e^{2\theta}-1)}$ with $\epsilon > 0$, the mechanism $M_t^f$ satisfies $(\alpha, \alpha\epsilon)$-RDP at time $t = 1$ and we have $\frac{\mathcal{E}_{\mathrm{OU}}(\theta,\rho,1)}{\mathcal{E}_{\mathrm{GM}}(\theta,\rho,1)} \leq \left(1 + \frac{d\Delta^2}{2\epsilon R^2}\right)^{-1}$.*

This result not only shows that the Ornstein-Uhlenbeck mechanism is uniformly better than the Gaussian mechanism for any level of privacy, but also shows that in this mechanism the error always stays bounded and can attain the same level of error as the Gaussian mechanism with optimal post-processing. To see this note that with the choices of parameters made in the second statement give $\mathcal{E}_{\mathrm{GM}}(\theta, \rho, 1) = \frac{d\Delta^2}{2\epsilon}$ and therefore $\mathcal{E}_{\mathrm{OU}}(\theta, \rho, 1) \leq \frac{d\Delta^2 R^2}{2\epsilon R^2 + d\Delta^2}$, which behaves like $O(R^2)$ with $\Delta$ constant and either $\epsilon \to 0$ or $d \to \infty$.

# 6 Conclusion

We have undertaken a systematic study of amplification by post-processing. Our results yield improvements over recent work on amplification by iteration, and introduce a new Ornstein-Uhlenbeck mechanism which is more accurate than the Gaussian mechanism. In the future it would be interesting to study applications of amplification by post-processing. One promising application is *Hierarchical Differential Privacy*, where information is released under increasingly strong privacy constraints (e.g. to a restricted group within a company, globally within a company, and finally to outside parties).

**Acknowledgements**   MG was partially supported by NSF grant CCF-1718220.

## Footnotes

[1]Rényi divergences do not belong to the family of Csiszár divergences.

[2]This divergence characterization of DP is due to [4].

[3]We take the convention $\tilde{\varepsilon} = \infty$ whenever $\delta = 0$, in which case the $(\gamma, \infty)$-Dobrushin condition is obtained with respect to the divergence $\mathsf{D}_\infty(\mu\|\nu) = \mu(\text{supp}(\mu) \setminus \text{supp}(\nu))$.

[4]The *blanket condition* is a necessary condition for LDP introduced in [3] to analyze privacy amplification by shuffling.

[5]Here we use the convention $\frac{0}{0} = 0$.

[6]That is, $\|\psi(x) - \psi(y)\| \le L\|x - y\|$ for any pair $x, y$.

[7]Here $\Pi_{\mathbb{K}}(x) = \arg\min_{y \in \mathbb{K}} \|x - y\|$ denotes the projection operator onto the convex set $\mathbb{K} \subseteq \mathbb{R}^d$.

[8]The details are not relevant here since we work directly with semigroups satisfying Assumption 1. We refer to [25] for details.

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
