[Supplementary Material · cr-supplementary.pdf]

# A  Proofs for Section 3 (Amplification From Uniform Mixing)

**Lemma 9.** *The implications in Figure 1 hold.*

*Proof.* That $(\gamma, \varepsilon)$-Dobrushin implies $\gamma$-Dobrushin follows directly from $\mathsf{D}_{e^\varepsilon}(K(x)\|K(x')) \leq \mathsf{TV}(K(x), K(x'))$.

To see that $\gamma$-Doeblin implies $\gamma$-Dobrushin we observe that the kernel of a $\gamma$-Doeblin operator must satisfy $\inf_x k(x, y) \geq (1 - \gamma)p_\omega(y)$ for any $y$. Thus, we can use the characterization of $\mathsf{TV}$ in terms of a minimum to get

$$\mathsf{TV}(K(x), K(x')) = 1 - \int (k(x, y) \wedge k(x', y))\lambda(dy) \leq 1 - (1 - \gamma)\int p_\omega(y)\lambda(dy) = \gamma \ .$$

Finally, to get the $\gamma$-Doeblin condition for an operator $K$ satisfying $\gamma$-ultra-mixing we recall from [12, Lemma 4.1] that for such an operator we have that $K(x) \geq (1 - \gamma)\tilde{\omega}K$ is satisfied for any probability distribution $\tilde{\omega}$ and $x \in \operatorname{supp}(\tilde{\omega})$. Thus, taking $\tilde{\omega}$ to have full support we obtain Doeblin's condition with $\omega = \tilde{\omega}K$. $\qquad\square$

For convenience, we split the proof of Theorem 1 into four separate statements, each corresponding to one of the claims in the theorem.

Recall that a Markov operator $K \in \mathcal{K}(\mathbb{X}, \mathbb{Y})$ is $\gamma$-*Dobrushin* if $\sup_{x,x'} \mathsf{TV}(K(x), K(x')) \leq \gamma$.

**Theorem 10.** *Let $M$ be an $(\varepsilon, \delta)$-DP mechanism. If $K$ is a $\gamma$-Dobrushin Markov operator, then the composition $K \circ M$ is $(\varepsilon, \gamma\delta)$-DP.*

*Proof.* This follows directly from the *strong Markov contraction lemma* established by **(author?)** [11] in the discrete case and by **(author?)** [12] in the general case (see also [26]). In particular, this lemma states that for any divergence $\mathsf{D}$ in the sense of Csiszár we have $\mathsf{D}(\mu K\|\nu K) \leq \gamma\mathsf{D}(\mu\|\nu)$. Letting $\mu = M(D)$ and $\nu = M(D')$ for some $D \simeq D'$ and applying this inequality to $\mathsf{D}_{e^\varepsilon}(\mu K\|\nu K)$ yields the result. $\qquad\square$

Next we prove amplification when $K$ is a $(\gamma, \varepsilon)$-Dobrushin operator. Recall that a Markov operator $K \in \mathcal{K}(\mathbb{X}, \mathbb{Y})$ is $(\gamma, \varepsilon)$-*Dobrushin* if $\sup_{x,x'} \mathsf{D}_{e^\varepsilon}(K(x)\|K(x')) \leq \gamma$. We will require the following technical lemmas in the proof of Theorem 13.

**Lemma 11.** *Let $\mu \perp \nu$ denote the fact $\operatorname{supp}(\mu) \cap \operatorname{supp}(\nu) = \emptyset$. If $K$ is $(\gamma, \varepsilon)$-Dobrushin, then we have*

$$\sup_{\mu \perp \nu} \mathsf{D}_{e^\varepsilon}(\mu K\|\nu K) \leq \gamma \ .$$

*Proof.* Note that the condition on $\gamma$ can be written as $\sup_{x,x'} \mathsf{D}_{e^\varepsilon}(\delta_x K\|\delta_{x'} K) \leq \gamma$. This shows that by hypothesis the condition already holds for the distributions $\delta_x \perp \delta_{x'}$ with $x \neq x'$. Thus, all we need to do is prove that these distributions are extremal for $\mathsf{D}_{e^\varepsilon}(\mu K\|\nu K)$ among all distributions with $\mu \perp \nu$. Let $\mu \perp \nu$ and define $U = \operatorname{supp}(\mu)$ and $V = \operatorname{supp}(\nu)$. Working in the discrete setting for simplicity, we can write $\mu = \sum_{x \in U} \mu(x)\delta_x$, with an equivalent expression for $\nu$. Now we use the joint convexity of $\mathsf{D}_{e^\varepsilon}$ to write

$$\mathsf{D}_{e^\varepsilon}(\mu K\|\nu K) \leq \sum_{x \in U} \mu(x)\mathsf{D}_{e^\varepsilon}(\delta_x K\|\nu K) \leq \sum_{x \in U}\sum_{x' \in V} \mu(x)\nu(x')\mathsf{D}_{e^\varepsilon}(\delta_x K\|\delta_{x'} K)$$

$$\leq \sup_{x \neq x'} D(\delta_x K\|\delta'_x K) \leq \gamma \ .$$

$\qquad\square$

**Lemma 12.** *Let $a \wedge b \triangleq \min\{a, b\}$. Then we have*

$$\mathsf{D}_{e^\varepsilon}(\mu\|\nu) = 1 - \int (p_\mu(x) \wedge e^\varepsilon p_\nu(x))\, \lambda(dx) \ .$$

*Proof.* Define $A = \{x : p_\mu(x) \le e^\varepsilon p_\nu(x)\}$ to be set of points where $\mu$ is dominated by $e^\varepsilon \nu$, and let $A^c$ denote its complementary. Then we have the identities

$$\int (p_\mu \wedge e^\varepsilon p_\nu) d\lambda = \int_A d\mu + e^\varepsilon \int_{A^c} d\nu \ ,$$

$$\int [p_\mu - e^\varepsilon p_\nu]_+ d\lambda = \int_{A^c} d\mu - e^\varepsilon \int_{A^c} d\nu \ .$$

Thus we obtain the desired result since

$$\mathsf{D}_{e^\varepsilon}(\mu\|\nu) + \int (p_\mu \wedge e^\varepsilon p_\nu) d\lambda = \int [p_\mu - e^\varepsilon p_\nu]_+ d\lambda + \int (p_\mu \wedge e^\varepsilon p_\nu) d\lambda = \int_{A^c} d\mu + \int_A d\mu = 1 \ .$$

$\square$

**Theorem 13.** *Let $M$ be an $(\varepsilon, \delta)$-DP mechanism and let $\varepsilon' = \log\left(1 + \frac{e^\varepsilon - 1}{\delta}\right)$. If $K$ is a $(\gamma, \varepsilon')$-Dobrushin Markov operator, then the composition $K \circ M$ is $(\varepsilon, \gamma\delta)$-DP.*

*Proof.* Fix $\mu = M(D)$ and $\nu = M(D')$ for some $D \simeq D'$ and let $\theta = \mathsf{D}_{e^\varepsilon}(\mu\|\nu) \le \delta$. We start by constructing overlapping mixture decompositions for $\mu$ and $\nu$ as follows. First, define the function $f = p_\mu \wedge e^\varepsilon p_\nu$ and let $\omega$ be the probability distribution with density $p_\omega = \frac{f}{\int f d\lambda} = \frac{f}{1-\theta}$, where we used Lemma 12. Now note that by construction we have the inequalities

$$p_\mu - (1-\theta) p_\omega = p_\mu - p_\mu \wedge e^\varepsilon p_\nu \ge 0 \ ,$$

$$p_\nu - \frac{1-\theta}{e^\varepsilon} p_\omega = p_\nu - p_\nu \wedge e^{-\varepsilon} p_\mu \ge 0 \ .$$

Assuming without loss of generality that $\mu \ne \nu$, these inequalities imply that we can construct probability distributions $\mu'$ and $\nu'$ such that

$$\mu = (1-\theta)\omega + \theta\mu' \ ,$$

$$\nu = \frac{1-\theta}{e^\varepsilon}\omega + \left(1 - \frac{1-\theta}{e^\varepsilon}\right)\nu' \ .$$

Now we observe that the distributions $\mu'$ and $\nu'$ defined in this way have disjoint support. To see this we first use the identity $p_\mu = (1-\theta)p_\omega + \theta p_{\mu'}$ to see that

$$p_{\mu'}(x) > 0 \equiv p_\mu(x) - (1-\theta)p_\omega(x) > 0 \equiv p_\mu(x) - p_\mu(x) \wedge e^\varepsilon p_\nu(x) > 0 \equiv p_\mu(x) > e^\varepsilon p_\nu(x) \ .$$

Thus we have $\mathrm{supp}(\mu') = \{x : p_\mu(x) > e^\varepsilon p_\nu(x)\}$. A similar argument applied to $p_\nu$ shows that on the other hand $\mathrm{supp}(\nu') = \{x : p_\mu(x) < e^\varepsilon p_\nu(x)\}$, and thus $\mu' \perp \nu'$.

Finally, we proceed to use the mixture decomposition of $\mu$ and $\nu$ and the condition $\mu' \perp \nu'$ to bound $\mathsf{D}_{e^\varepsilon}(\mu K\|\nu K)$ as follows. By using the mixture decompositions we get

$$\mu - e^\varepsilon \nu = \theta\mu' - e^\varepsilon\left(1 - \frac{1-\theta}{e^\varepsilon}\right)\nu' = \theta(\mu' - e^{\tilde\varepsilon}\nu') \ ,$$

where $\tilde\varepsilon = \log\left(1 + \frac{e^\varepsilon - 1}{\theta}\right) \ge \varepsilon'$. Thus, applying the definition of $\mathsf{D}_{e^\varepsilon}$, using the linearity of Markov operators, and the monotonicity $\mathsf{D}_{e^{\tilde\varepsilon}} \le \mathsf{D}_{e^{\varepsilon'}}$ we obtain the bound:

$$\mathsf{D}_{e^\varepsilon}(\mu K\|\nu K) = \theta\mathsf{D}_{e^{\tilde\varepsilon}}(\mu' K\|\nu' K) \le \theta\mathsf{D}_{e^{\varepsilon'}}(\mu' K\|\nu' K) \le \gamma\theta = \gamma\mathsf{D}_{e^{\varepsilon'}}(\mu\|\nu) \ ,$$

where the last inequality follows from Lemma 11. $\square$

Recall that a Markov operator $K \in \mathcal{K}(\mathbb{X}, \mathbb{Y})$ is $\gamma$-*Doeblin* if there exists a distribution $\omega \in \mathcal{P}(\mathbb{Y})$ such that $K(x) \ge (1-\gamma)\omega$ for all $x \in \mathbb{X}$. The proof of amplification for $\gamma$-Doeblin operators further leverages overlapping mixture decompositions like the one used in Theorem 13, but this time the mixture arises at the level of the kernel itself.

**Theorem 14.** *Let $M$ be an $(\varepsilon, \delta)$-DP mechanism. If $K$ is a $\gamma$-Doeblin Markov operator, then the composition $K \circ M$ is $(\varepsilon', \delta')$-DP with $\varepsilon' = \log(1 + \gamma(e^\varepsilon - 1))$ and $\delta' = \gamma\left(1 - e^{\varepsilon'-\varepsilon}(1-\delta)\right)$.*

*Proof.* Fix $\mu = M(D)$ and $\nu = M(D')$ for some $D \simeq D'$. Let $\omega$ be a witness that $K$ is $\gamma$-Doeblin and let $K_\omega$ be the constant Markov operator given by $K_\omega(x) = \omega$ for all $x$. Doeblin's condition $K(x) \geq (1-\gamma)\omega = (1-\gamma)K_\omega(x)$ implies that the following is again a Markov operator:

$$\tilde{K} = \frac{K - (1-\gamma)K_\omega}{\gamma} \ .$$

Thus, we can write $K$ as the mixture $K = (1-\gamma)K_\omega + \gamma\tilde{K}$ and then use the *advanced joint convexity* property of $\mathsf{D}_{e^{\varepsilon'}}$ [2, Theorem 2] with $\varepsilon' = \log(1 + \gamma(e^\varepsilon - 1))$ to obtain the following:

$$\begin{aligned}
\mathsf{D}_{e^{\varepsilon'}}(\mu K \| \nu K) &= \mathsf{D}_{e^{\varepsilon'}}((1-\gamma)\omega + \gamma\mu\tilde{K} \| (1-\gamma)\omega + \gamma\nu\tilde{K}) \\
&= \gamma\mathsf{D}_{e^\varepsilon}(\mu\tilde{K} \| (1-\beta)\omega + \beta\nu\tilde{K}) \\
&\leq \gamma\left((1-\beta)\mathsf{D}_{e^\varepsilon}(\mu\tilde{K} \| \omega) + \beta\mathsf{D}_{e^\varepsilon}(\mu\tilde{K} \| \nu\tilde{K})\right) \ ,
\end{aligned}$$

where $\beta = e^{\varepsilon' - \varepsilon}$. Finally, using the immediate bounds $\mathsf{D}_{e^\varepsilon}(\mu\tilde{K} \| \nu\tilde{K}) \leq \mathsf{D}_{e^\varepsilon}(\mu \| \nu)$ and $\mathsf{D}_{e^\varepsilon}(\mu\tilde{K} \| \omega) \leq 1$, we get

$$\mathsf{D}_{e^{\varepsilon'}}(\mu K \| \nu K) \leq \gamma(1 - e^{\varepsilon' - \varepsilon} + e^{\varepsilon' - \varepsilon}\delta) \ .$$

$\square$

Our last amplification result applies to operators satisfying the ultra-mixing condition of **(author?)** [12]. We say that a Markov operator $K \in \mathcal{K}(\mathbb{X}, \mathbb{Y})$ is $\gamma$-*ultra-mixing* if for all $x, x' \in \mathbb{X}$ we have $K(x) \ll K(x')$ and $\frac{dK(x)}{dK(x')} \geq 1 - \gamma$. The proof strategy is based on the ideas from the previous proof, although in this case the argument is slightly more technical as it involves a strengthening of the Doeblin condition implied by ultra-mixing that only holds under a specific support.

**Theorem 15.** *Let $M$ be an $(\varepsilon, \delta)$-DP mechanism. If $K$ is a $\gamma$-ultra-mixing Markov operator, then the composition $K \circ M$ is $(\varepsilon', \delta')$-DP with $\varepsilon' = \log(1 + \gamma(e^\varepsilon - 1))$ and $\delta' = \gamma\delta e^{\varepsilon' - \varepsilon}$.*

*Proof.* Fix $\mu = M(D)$ and $\nu = M(D')$ for some $D \simeq D'$. The proof follows a similar strategy as the one used in Theorem 14, but coupled with the following consequence of the ultra-mixing property: for any probability distribution $\omega$ and $x \in \mathrm{supp}(\omega)$ we have $K(x) \geq (1-\gamma)\omega K$ [12, Lemma 4.1]. We use this property to construct a collection of mixture decompositions for $K$ as follows. Let $\alpha \in (0,1)$ and take $\tilde{\omega} = (1-\alpha)\mu + \alpha\nu$ and $\omega = \tilde{\omega}K$. By the ultra-mixing condition and the argument used in the proof of Theorem 14, we can show that

$$\tilde{K} = \frac{K - (1-\gamma)K_\omega}{\gamma}$$

is a Markov operator from $\mathrm{supp}(\mu) \cup \mathrm{supp}(\nu)$ into $\mathbb{X}$. Here $K_\omega$ is the constant Markov operator $K_\omega(x) = \omega$. Furthermore, the expression for $\tilde{K}$ and the definition of $\omega$ imply that

$$\tilde{\omega}\tilde{K} = \frac{\tilde{\omega}K - (1-\gamma)\tilde{\omega}K_\omega}{\gamma} = \omega \ . \tag{5}$$

Now note that the mixture decompositions $\mu K = (1-\gamma)\omega + \gamma\mu\tilde{K}$ and $\nu K = (1-\gamma)\omega + \gamma\nu\tilde{K}$ and the *advanced joint convexity* property of $\mathsf{D}_{e^{\varepsilon'}}$ [2, Theorem 2] with $\varepsilon' = \log(1 + \gamma(e^\varepsilon - 1))$ yield

$$\begin{aligned}
\mathsf{D}_{e^{\varepsilon'}}(\mu K \| \nu K) &\leq \gamma\left((1-\beta)\mathsf{D}_{e^\varepsilon}(\mu\tilde{K} \| \omega) + \beta\mathsf{D}_{e^\varepsilon}(\mu\tilde{K} \| \nu\tilde{K})\right) \\
&\leq \gamma\left((1-\beta)\mathsf{D}_{e^\varepsilon}(\mu\tilde{K} \| \omega) + \beta\mathsf{D}_{e^\varepsilon}(\mu \| \nu)\right) \\
&\leq \gamma\left((1-\beta)\mathsf{D}_{e^\varepsilon}(\mu\tilde{K} \| \omega) + \beta\delta\right) \ ,
\end{aligned}$$

where $\beta = e^{\varepsilon' - \varepsilon}$. Using (5) we can expand the remaining divergence above as follows:

$$\mathsf{D}_{e^\varepsilon}(\mu\tilde{K} \| \omega) = \mathsf{D}_{e^\varepsilon}(\mu\tilde{K} \| \tilde{\omega}\tilde{K}) \leq \mathsf{D}_{e^\varepsilon}(\mu \| \tilde{\omega}) \leq \alpha\mathsf{D}_{e^\varepsilon}(\mu \| \nu) \leq \alpha\delta \ ,$$

where we used the definition of $\tilde{\omega}$ and joint convexity. Since $\alpha$ was arbitrary, we can now take the limit $\alpha \to 0$ to obtain the bound $\mathsf{D}_{e^{\varepsilon'}}(\mu K \| \nu K) \leq \gamma\delta e^{\varepsilon' - \varepsilon}$. $\square$

*Proof of Theorem 1.* It follows from Theorems 10, 13, 14 and 15. $\square$

# B Proofs for Section 4 (Amplification From Couplings)

**Lemma 16.** *The transport operator $H_\pi$ with $\pi \in \mathcal{C}(\mu, \nu)$ satisfies $\mu H_\pi = \nu$.*

*Proof.* Take an arbitrary event $E$ and note that:

$$(\mu H_\pi)(E) = \int_\mathbb{X} H_\pi(x)(E)\mu(dx) = \int_\mathbb{X}\int_E h_\pi(x,y)\mu(dx)\lambda(dy) = \int_\mathbb{X}\int_E \frac{p_\pi(x,y)}{p_\mu(x)}\mu(dx)\lambda(dy)$$

$$= \int_\mathbb{X}\int_E p_\pi(x,y)\lambda(dx)\lambda(dy) = \int_E p_\nu(y)\lambda(dy) = \nu(E) \ ,$$

where we used the coupling property $\int_\mathbb{X} p_\pi(x,y)\lambda(dx) = p_\nu(y)$. $\qquad\square$

**Theorem 2.** *Let $\alpha \geq 1$, $\mu, \nu \in \mathcal{P}(\mathbb{X})$ and $K \in \mathcal{K}(\mathbb{X}, \mathbb{Y})$. For any distribution $\omega \in \mathcal{P}(\mathbb{X})$ and coupling $\pi \in \mathcal{C}(\omega, \mu)$ we have*

$$\mathsf{R}_\alpha(\mu K \| \nu K) \leq \mathsf{R}_\alpha(\omega \| \nu) + \sup_{x \in \mathrm{supp}(\nu)} \mathsf{R}_\alpha((H_\pi K)(x) \| K(x)) \ . \tag{1}$$

*Proof.* Let $\omega \in \mathcal{P}(\mathbb{X})$ and $\pi \in \mathcal{C}(\omega, \mu)$ be as in the statement, and let $\pi' = C(\mu, \omega)$. Note that taking $H_\pi$ and $H_{\pi'}$ to be the corresponding transport operators we have $\mu = \mu H_{\pi'} H_\pi = \omega H_\pi$. Now, given a $\lambda \in \mathcal{P}(\mathbb{X} \times \mathbb{X})$ let $\Pi_2(\lambda) = \int \lambda(dx, \cdot)$ denote the marginal of $\lambda$ on the second coordinate. In particular, if $\mu \otimes K$ denotes the joint distribution of $\mu$ and $\mu K$, then we have $\Pi_2(\mu \otimes K) = \mu K$. Thus, by the data processing inequality we have

$$\mathsf{R}_\alpha(\mu K \| \nu K) = \mathsf{R}_\alpha(\omega H_\pi K \| \nu K) = \mathsf{R}_\alpha(\Pi_2(\omega \otimes H_\pi K) \| \Pi_2(\nu \otimes K)) \leq \mathsf{R}_\alpha(\omega \otimes H_\pi K \| \nu \otimes K) \ .$$

The final step is to expand the RHS of the derivation above as follows:

$$e^{(\alpha-1)\mathsf{R}_\alpha(\omega \otimes H_\pi K \| \nu \otimes K)} = \iint \left(\frac{d(\omega \otimes H_\pi K)}{d(\nu \otimes K)}\right)^\alpha \nu(dx)K(x, dy)$$

$$= \iint \left(\frac{p_\omega(x)\int h_\pi(x, dz)k(z, y)}{p_\nu(x)k(x, y)}\right)^\alpha \nu(dx)K(x, dy)$$

$$= \iint \left(\frac{p_\omega(x)}{p_\nu(x)}\right)^\alpha \left(\frac{\int h_\pi(x, dz)k(z, y)}{k(x, y)}\right)^\alpha \nu(dx)K(x, dy)$$

$$\leq \left(\int \left(\frac{p_\omega(x)}{p_\nu(x)}\right)^\alpha \nu(dx)\right)\left(\sup_x \int \left(\frac{\int h_\pi(x, dz)k(z, y)}{k(x, y)}\right)^\alpha K(x, dy)\right)$$

$$= e^{(\alpha-1)\mathsf{R}_\alpha(\omega \| \nu)} \cdot e^{(\alpha-1)\sup_x \mathsf{R}_\alpha((H_\pi K)(x) \| K(x))} \ ,$$

where the supremums are taken with respect to $x \in \mathrm{supp}(\nu)$. $\qquad\square$

**Lemma 3.** *Let $M(D) = \mathrm{Lap}(f(D), \lambda_1)$ for some function $f : \mathbb{D} \to \mathbb{R}$ with global $L_1$-sensitivity $\Delta$ and let the Markov operator $K$ be given by $K(x) = \mathrm{Lap}(x, \lambda_2)$. The post-processed mechanism $(K \circ M)$ does not achieve $(\varepsilon, 0)$-DP for any $\varepsilon < \frac{\Delta}{\max\{\lambda_1, \lambda_2\}}$. Note that $M$ achieves $(\frac{\Delta}{\lambda_1}, 0)$-DP and $K(f(D))$ achieves $(\frac{\Delta}{\lambda_2}, 0)$-DP.*

*Proof.* This can be shown by directly analyzing the distribution arising from the sum of two independent laplace variables. Let $Lap2(\lambda_1, \lambda_2)$ denote this distribution. In the following equations, we assume $x > 0$. Due to symmetry around the origin, densities at negative values can be found by looking instead at the corresponding positive location.

$$Lap2(x; \lambda_1, \lambda_2) = \int_{-\infty}^{\infty} \frac{1}{2\lambda_1} \exp\left(-\frac{|x-t|}{\lambda_1}\right) \frac{1}{2\lambda_2} \exp\left(-\frac{|t|}{\lambda_2}\right) dt$$

$$= \frac{1}{4\lambda_1\lambda_2} \int_{-\infty}^{\infty} \exp\left(-\frac{\lambda_2|x-t| + \lambda_1|t|}{\lambda_1\lambda_2}\right) dt$$

$$= \frac{1}{4\lambda_1\lambda_2} \left( \int_{-\infty}^{0} e^{-\frac{\lambda_2(x-t)-\lambda_1 t}{\lambda_1\lambda_2}} dt + \int_{0}^{x} e^{-\frac{\lambda_2(x-t)+\lambda_1 t}{\lambda_1\lambda_2}} dt + \int_{x}^{\infty} e^{-\frac{-\lambda_2(x-t)+\lambda_1 t}{\lambda_1\lambda_2}} dt \right)$$

$$= \frac{1}{4\lambda_1\lambda_2} \left( \int_{-\infty}^{0} e^{-\frac{\lambda_2 x - (\lambda_1+\lambda_2)t}{\lambda_1\lambda_2}} dt + \int_{0}^{x} e^{-\frac{\lambda_2 x + (\lambda_1-\lambda_2)t}{\lambda_1\lambda_2}} dt + \int_{x}^{\infty} e^{-\frac{-\lambda_2 x + (\lambda_1+\lambda_2)t}{\lambda_1\lambda_2}} dt \right)$$

$$= \frac{1}{4\lambda_1\lambda_2} \left( \frac{e^{-\frac{\lambda_2 x - (\lambda_1+\lambda_2)t}{\lambda_1\lambda_2}}}{(\lambda_1+\lambda_2)/\lambda_1\lambda_2}\Big|_{t=-\infty}^{t=0} + \int_{0}^{x} e^{-\frac{\lambda_2 x + (\lambda_1-\lambda_2)t}{\lambda_1\lambda_2}} dt + \frac{e^{-\frac{-\lambda_2 x + (\lambda_1+\lambda_2)t}{\lambda_1\lambda_2}}}{(\lambda_1+\lambda_2)/\lambda_1\lambda_2}\Big|_{t=x}^{t=\infty} \right)$$

The integration on the middle term varies between the cases $\lambda_1 = \lambda_2$ and $\lambda_1 \neq \lambda_2$. Finishing this derivation and replacing $x$ with $|x|$ to account for both positive and negative values, we get a complete expression for our $Lap2(\lambda_1, \lambda_2)$ density.

$$Lap2(x; \lambda_1, \lambda_2) = \begin{cases} \frac{1}{4}\left( (\frac{1}{\lambda_1+\lambda_2} + \frac{1}{\lambda_1-\lambda_2})e^{-\frac{|x|}{\lambda_1}} + (\frac{1}{\lambda_1+\lambda_2} - \frac{1}{\lambda_1-\lambda_2})e^{-\frac{|x|}{\lambda_2}} \right) & \text{if } \lambda_1 \neq \lambda_2 \ , \\ \frac{1}{4\lambda_1^2}e^{-\frac{|x|}{\lambda_1}}(\lambda_1 + |x|) & \text{if } \lambda_1 = \lambda_2 \ . \end{cases} \quad (6)$$

To finish this lemma, we need to derive the best $(\epsilon, 0)$-DP guarantee offered by adding noise from $Lap2(\lambda_1, \lambda_2)$. From the post-processing property of DP and the commutivity of additive mechanisms, we know this guarantee is upper-bounded by $\Delta/\max\{\lambda_1, \lambda_2\}$. A direct computation of $\lim_{x\to\infty} \log(Lap2(x; \lambda_1, \lambda_2)/Lap2(x+\Delta; \lambda_1, \lambda_2))$ results in $\Delta/\max\{\lambda_1, \lambda_2\}$ in both cases of equation (6). This arises from the limit depending entirely on the dominating term with the largest exponent. Therefore, this lower-bounds the privacy guarantee by the same value. Thus we can conclude this is the exact level of $(\epsilon, 0)$-DP offered by this mechanism.

$\square$

**Theorem 4.** *Let $\alpha \geq 1$, $\mu, \nu \in \mathcal{P}(\mathbb{R}^d)$ and let $\mathbb{K} \subseteq \mathbb{R}^d$ be a convex set. Suppose $K_1, \ldots, K_r \in \mathcal{K}(\mathbb{R}^d, \mathbb{R}^d)$ are Markov operators where $Y_i \sim K_i(x)$ is obtained as[9] $Y_i = \Pi_{\mathbb{K}}(\psi_i(x) + Z_i)$ with $Z_i \sim \mathcal{N}(0, \sigma^2 I)$, where the maps $\psi_i : \mathbb{K} \to \mathbb{R}^d$ are L-Lipschitz for all $i \in [r]$. For any $\mu_0, \mu_1, \ldots, \mu_r \in \mathcal{P}(\mathbb{R}^d)$ with $\mu_0 = \mu$ and $\mu_r = \nu$ we have*

$$\mathsf{R}_\alpha(\mu K_1 \cdots K_r \| \nu K_1 \cdots K_r) \leq \frac{\alpha L^2}{2\sigma^2} \sum_{i=1}^{r} L^{2(r-i)} \mathsf{W}_\infty(\mu_i, \mu_{i-1})^2 \ . \quad (2)$$

*Furthermore, if $L \leq 1$ and $\mathsf{W}_\infty(\mu, \nu) = \Delta$, then*

$$\mathsf{R}_\alpha(\mu K_1 \cdots K_r \| \nu K_1 \cdots K_r) \leq \frac{\alpha \Delta^2 L^{r+1}}{2r\sigma^2} \ . \quad (3)$$

The proof of Theorem 4 relies on the following technical lemma about the effect of a projected Lipschitz Gaussian operator on the $\infty$-Wasserstein distance between two distributions.

**Lemma 17.** *Let $\mathbb{K} \subseteq \mathbb{R}^d$ be a convex set and $\psi : \mathbb{K} \to \mathbb{R}^d$ be L-Lipschitz. Suppose $K \in \mathcal{K}(\mathbb{R}^d, \mathbb{R}^d)$ is a Markov operator where $Y \sim K(x)$ is obtained as $Y = \Pi_{\mathbb{K}}(\psi(x) + Z)$ with $Z \sim \mathcal{N}(0, \sigma^2 I)$. Then, for any $\mu, \nu \in \mathcal{P}(\mathbb{R}^d)$ we have $\mathsf{W}_\infty(\mu K, \nu K) \leq L \mathsf{W}_\infty(\mu, \nu)$.*

*Proof.* Let $\pi \in \mathcal{C}(\mu, \nu)$ be a witness of $\mathsf{W}_\infty(\mu, \nu) = \Delta$. We construct a witness of $\mathsf{W}_\infty(\mu K, \nu K) \leq L\Delta$ as follows: sample $(X, X') \sim \pi$ and $Z \sim \mathcal{N}(0, \sigma^2 I)$ and then let $Y = \Pi_{\mathbb{K}}(\psi(X) + Z)$ and $Y' = \Pi_{\mathbb{K}}(\psi(X') + Z)$. It is clear from the construction that $\mathsf{Law}((Y, Y')) \in \mathcal{C}(\mu K, \nu K)$.

Furthermore, by the Lipschitz assumption on $\psi$ and that fact that the map $\Pi_{\mathbb{K}}$ is contractive, the following holds almost surely:

$$\|Y - Y'\| \le \|\psi(X) - \psi(X')\| \le L\|X - X'\| \le L\Delta \ .$$

$\square$

*Proof of Theorem 4.* We prove (2) by induction on $r$. For the base case $r = 1$ we apply Theorem 2 with $\omega = \nu$ and a coupling $\pi \in \mathcal{C}(\nu, \mu)$ witnessing that $\mathsf{W}_\infty(\mu, \nu) = \Delta$. This choice of coupling guarantees that for any $x \in \mathrm{supp}(\nu)$ we have $\mathrm{supp}(H_\pi(x)) \subseteq B_\Delta(x)$, where $B_\Delta(x)$ is the ball of radius $\Delta$ around $x$. Note also that $(H_\pi K_1)(x) = H_\pi(x)K_1$. Thus, from (1) we obtain, using Hölder's inequality and the monotonicity of the logarithm, that:

$$\mathsf{R}_\alpha(\mu K_1 \| \nu K_1) \le \sup_{x \in \mathrm{supp}(\nu)} \mathsf{R}_\alpha((H_\pi K_1)(x)\|K_1(x)) \le \sup_{x \in \mathrm{supp}(\nu)} \sup_{y \in \mathrm{supp}(H_\pi(x))} \mathsf{R}_\alpha(K_1(y)\|K_1(x))$$

$$\le \sup_{\|x-y\| \le \Delta} \mathsf{R}_\alpha(K_1(y)\|K_1(x)) \ .$$

Now note that the Markov operator $K_1$ can be obtained by post-processing $\tilde{K}_1(x) = \mathcal{N}(\psi_1(x), \sigma^2 I)$ with the projection $\Pi_{\mathbb{K}}$. Thus, by the data processing inequality we obtain

$$\sup_{\|x-y\| \le \Delta} \mathsf{R}_\alpha(K_1(y)\|K_1(x)) \le \sup_{\|x-y\| \le \Delta} \mathsf{R}_\alpha(\tilde{K}_1(y)\|\tilde{K}_1(x))$$

$$= \sup_{\|x-y\| \le \Delta} \frac{\alpha\|\psi_1(x) - \psi_1(y)\|^2}{2\sigma^2} \le \frac{\alpha\Delta^2 L^2}{2\sigma^2} \ .$$

For the inductive case we suppose that (2) holds for some $r \ge 1$ and consider the case $r + 1$, in which we need to bound $\mathsf{R}_\alpha(\mu K_1 \cdots K_{r+1} \| \nu K_1 \cdots K_{r+1})$. Let $\mu_0, \mu_1, \ldots, \mu_{r+1}$ be a sequence of distributions with $\mu_0 = \mu$ and $\mu_{r+1} = \nu$. Applying (1) with $\omega = \mu_1 K_1 \cdots K_r$ and some coupling $\pi \in \mathcal{C}(\mu_1 K_1 \cdots K_r, \mu K_1 \cdots K_r)$ we have

$$\mathsf{R}_\alpha(\mu K_1 \cdots K_{r+1} \| \nu K_1 \cdots K_{r+1}) \le \mathsf{R}_\alpha(\mu_1 K_1 \cdots K_r \| \nu K_1 \cdots K_r)$$
$$+ \sup_{x \in \mathrm{supp}(\nu K_1 \cdots K_r)} \mathsf{R}_\alpha((H_\pi K_{r+1})(x)\|K_{r+1}(x)) \ .$$

By the inductive hypothesis, the first term in the RHS above can be bounded as follows:

$$\mathsf{R}_\alpha(\mu_1 K_1 \cdots K_r \| \nu K_1 \cdots K_r) \le \frac{\alpha L^2}{2\sigma^2} \sum_{i=1}^{r} L^{2(r-i)}\mathsf{W}_\infty(\mu_{i+1}, \mu_i)^2$$

$$= \frac{\alpha L^2}{2\sigma^2} \sum_{i=2}^{r+1} L^{2(r+1-i)}\mathsf{W}_\infty(\mu_i, \mu_{i-1})^2 \ .$$

To bound the second term we assume the coupling $\pi$ is a witness of $\mathsf{W}_\infty(\mu_1 K_1 \cdots K_r, \mu K_1 \cdots K_r) = \Delta'$, in which case a similar argument to the one we used in the base case yields:

$$\sup_x \mathsf{R}_\alpha((H_\pi K_{r+1})(x)\|K_{r+1}(x)) \le \sup_x \sup_{y \in \mathrm{supp}(H_\pi(x))} \mathsf{R}_\alpha(K_{r+1}(y)\|K_{r+1}(x))$$

$$\le \sup_{\|x-y\| \le \Delta'} \mathsf{R}_\alpha(K_{r+1}(y)\|K_{r+1}(x))$$

$$\le \frac{\alpha\Delta'^2 L^2}{2\sigma^2} \le \frac{\alpha L^{2r+2}\mathsf{W}_\infty(\mu_1, \mu)^2}{2\sigma^2} \ ,$$

where the last inequality follows from Lemma 17. Plugging the last three inequalities together we finally obtain

$$\mathsf{R}_\alpha(\mu K_1 \cdots K_{r+1} \| \nu K_1 \cdots K_{r+1}) \le \frac{\alpha L^{2r+2}\mathsf{W}_\infty(\mu_1, \mu_0)^2}{2\sigma^2} + \frac{\alpha L^2}{2\sigma^2} \sum_{i=2}^{r+1} L^{2(r+1-i)}\mathsf{W}_\infty(\mu_i, \mu_{i-1})^2$$

$$= \frac{\alpha L^2}{2\sigma^2} \sum_{i=1}^{r+1} L^{2(r+1-i)}\mathsf{W}_\infty(\mu_i, \mu_{i-1})^2 \ .$$

When $L \leq 1$, we can obtain (3) from (2) as follows. First, construct a sequence of distributions $\mu_0, \ldots, \mu_r$ such that $\Delta_i \triangleq W_\infty(\mu_i, \mu_{i-1}) = \Delta_0 L^i$ for $i \in [r]$, where $\Delta_0 = \frac{\Delta}{L} \frac{1-L}{1-L^r}$ is a normalization constant chosen such that $\sum_{i \in [r]} \Delta_i = \Delta$. With this choice plugged into (2) we obtain

$$R_\alpha(\mu K_1 \cdots K_r \| \nu K_1 \cdots K_r) \leq \frac{\alpha L^2}{2\sigma^2} r \Delta_0^2 L^{2r} = \frac{\alpha \Delta^2 L^{r+1} r}{2\sigma^2} \left( \frac{L^{-\frac{1}{2}} - L^{\frac{1}{2}}}{L^{-\frac{r}{2}} - L^{\frac{r}{2}}} \right)^2 = \frac{\alpha \Delta^2 L^{r+1} r}{2\sigma^2} \phi(L)^2 \ .$$

Now we note the function $\phi(L)$ defined above is increasing in $[0, 1]$ and furthermore $\lim_{L \to 1} \phi(L) = \frac{1}{r}$, which can be checked by applying L'Hôpital's rule twice. Thus, we can plug the inequality $\phi(L) \leq \frac{1}{r}$ above to obtain (3).

But we still need to show that a sequence $\mu_0, \ldots, \mu_r$ with $\Delta_i$ as above exists. To construct such a sequence we let $\pi \in \mathcal{C}(\mu, \nu)$ be a witness of $W_\infty(\mu, \nu) = \Delta$, take random variables $(X, X') \sim \pi$, and define $\mu_i = \mathsf{Law}((1 - \theta_i)X + \theta_i X')$ with $\theta_i = \frac{\Delta_0}{\Delta} \sum_{j=1}^{i} L^j = \frac{1-L^i}{1-L^r}$. Clearly we get $\mu_0 = \mathsf{Law}(X) = \mu$ and $\mu_r = \mathsf{Law}(X') = \nu$.

To see that $W_\infty(\mu_i, \mu_{i-1}) \leq \Delta_0 L^i$ we construct a coupling between $\mu_i$ and $\mu_{i-1}$ as follows: sample $(X, X') \sim \pi$ and let $Y = (1 - \theta_i)X + \theta_i X'$ and $Y' = (1 - \theta_{i-1})X + \theta_{i-1} X'$. Clearly we have $\mathsf{Law}((Y, Y')) \in \mathcal{C}(\mu_i, \mu_{i-1})$. Furthermore, with probability one the following holds:

$$\|Y - Y'\| = \|(\theta_{i-1} - \theta_i)X - (\theta_{i-1} - \theta_i)X'\| = \frac{\Delta_0}{\Delta} L^i \|X - X'\| \leq \Delta_0 L^i \ ,$$

where the last inequality uses that $\pi$ is a witness of $W_\infty(\mu, \nu) \leq \Delta$. This concludes the proof. $\square$

**Theorem 5.** *Let $\ell : \mathbb{K} \times \mathbb{D} \to \mathbb{R}$ be a C-Lipschitz, $\beta$-smooth, $\rho$-strongly convex loss function. If $\eta \leq \frac{2}{\beta+\rho}$, then $\mathrm{NoisyProjSGD}(D, \ell, \eta, \sigma, \xi_0)$ satisfies $(\alpha, \alpha\epsilon_i)$-RDP at index $i$, where $\epsilon_n = \frac{2C^2}{\sigma^2}$ and $\epsilon_i = \frac{2C^2}{(n-i)\sigma^2}(1 - \frac{2\eta\beta\rho}{\beta+\rho})^{\frac{n-i+1}{2}}$ for $1 \leq i \leq n-1$.*

To prove Theorem 5 we will use the following well-known fact about convex optimization: gradient iterations on a strongly convex function are strict contractions. The lemma below provides an expression for the contraction coefficient.

**Lemma 18.** *Let $\mathbb{K} \subseteq \mathbb{R}^d$ be a convex set and suppose the function $f : \mathbb{K} \to \mathbb{R}$ is $\beta$-smooth and $\rho$-strongly convex. If $\eta \leq \frac{2}{\beta+\rho}$, then the map $\psi(x) = x - \eta\nabla f(x)$ is L-Lipschitz on $\mathbb{K}$ with $L = \sqrt{1 - \frac{2\eta\beta\rho}{\beta+\rho}} < 1$.*

*Proof.* This follows from a standard calculation in convex optimization; see e.g. [7, Theorem 3.12]. We reproduce the proof here for completeness. Recall from [7, Lemma 3.11] that if a function $f$ is $\beta$-smooth and $\rho$-strongly convex, then for any $x, y \in \mathbb{K}$ we have

$$\frac{\beta\rho}{\beta+\rho}\|x - y\|^2 + \frac{1}{\beta+\rho}\|\nabla f(x) - \nabla f(y)\|^2 \leq \langle \nabla f(x) - \nabla f(y), x - y \rangle \ .$$

Using this inequality, one can show the following:

$$
\begin{aligned}
\|\psi(x) - \psi(y)\|^2 &= \|(x - \eta\nabla f(x)) - (y - \eta\nabla f(y))\|^2 \\
&= \|x - y\|^2 + \eta^2\|\nabla f(x) - \nabla f(y)\|^2 - 2\eta\langle \nabla f(x) - \nabla f(y), x - y \rangle \\
&\leq \left(1 - \frac{2\eta\beta\rho}{\beta+\rho}\right)\|x - y\|^2 + \eta\left(\eta - \frac{2}{\beta+\rho}\right)\|\nabla f(x) - \nabla f(y)\|^2 \\
&\leq \left(1 - \frac{2\eta\beta\rho}{\beta+\rho}\right)\|x - y\|^2 \ ,
\end{aligned}
$$

where the last inequality uses our assumption on $\eta$. $\square$

*Proof of Theorem 5.* Fix $1 \leq i \leq n-1$ and let $D \simeq D'$ be two datasets differing on the $i$th coordinate. Let $\xi \triangleq \xi_{i-1} \in \mathcal{P}(\mathbb{R}^d)$ represent the distribution of $x_{i-1}$ in the execution of Algorithm 1 with input $D$. Since $D$ and $D'$ differ only on the $i$th coordinate, the distribution of $x_{i-1}$ on input $D'$ is also $\xi$. Now let $\psi_0(x) = x - \eta\nabla_x\ell(x, z_i)$, $\psi_0'(x) = x - \eta\nabla_x\ell(x, z_i')$, and $\psi_j(x) = x - \eta\nabla_x\ell(x, z_{i+j})$ for $j \in [r]$

with $r = n - i$. Defining the Markov operators $K_j$, $j \in \{0, \ldots, r\}$, where $Y_j \sim K_j(x)$ is given by $K_j(x) = \Pi_\mathbb{K}(\psi_j(x) + Z)$ with $Z \sim \mathcal{N}(0, \eta^2\sigma^2 I)$, we immediately obtain that the distribution of the output $x_n$ of NoisyProjSGD$(D, \ell, \eta, \sigma)$ can be written as $\xi K_0 K_1 \cdots K_r$. Similarly, the distribution of the output of NoisyProjSGD$(D', \ell, \eta, \sigma)$ can be written as $\xi K_0' K_1 \cdots K_r$, where $K_0'(x) = \mathcal{N}(\psi_0'(x), \eta^2\sigma^2 I)$. Therefore, to obtain the Rényi differential privacy of NoisyProjSGD$(D, \ell, \eta, \sigma)$ at index $i$ we need to bound $\mathsf{R}_\alpha(\xi K_0 K_1 \cdots K_r \| \xi K_0' K_1 \cdots K_r)$.

With the goal to apply Theorem 4, we first define $\mu = \xi K_0$ and $\nu = \xi K_0'$ and use the Lipschitz assumption on $\ell$ to conclude that $\mathsf{W}_\infty(\mu, \nu) \leq 2\eta C$. Indeed, consider the coupling $\pi \in \mathcal{C}(\mu, \nu)$ obtained by sampling $(Y, Y') \sim \pi$ as follows: sample $X \sim \xi$ and $Z \sim \mathcal{N}(0, \eta^2\sigma^2 I)$, and then let $Y = \Pi_\mathbb{K}(\psi_0(X) + Z)$ and $Y' = \Pi_\mathbb{K}(\psi_0'(X) + Z)$. Now, since $\ell(\cdot, z_i)$ and $\ell(\cdot, z_i')$ are both $C$-Lipschitz and $\Pi_\mathbb{K}$ is contractive, we see that the following holds almost surely under $\pi$:

$$
\begin{aligned}
\|Y - Y'\| &\leq \|\psi_0(X) - \psi_0'(X)\| = \eta\|\nabla_x\ell(X, z_i) - \nabla_x\ell(X, z_i')\| \\
&\leq \eta\left(\|\nabla_x\ell(X, z_i)\| + \|\nabla_x\ell(X, z_i)\|\right) \leq 2\eta C .
\end{aligned}
$$

Thus, $\mathsf{W}_\infty(\mu, \nu) \leq 2\eta C$ as claimed.

Next we note that the assumption $\eta \leq \frac{2}{\beta + \rho}$ together with Lemma 18 imply that $\psi_j$, $j \in [r]$, are all $L$-Lipschitz with $L = \sqrt{1 - \frac{2\eta\beta\rho}{\beta+\rho}} < 1$. Thus we can apply Theorem 4 with $\Delta = 2\eta C$ to obtain

$$
\mathsf{R}_\alpha(\xi K_0 K_1 \cdots K_r \| \xi K_0' K_1 \cdots K_r) \leq \frac{2\alpha\eta^2 C^2 L^{n-i+1}}{(n-i)\eta^2\sigma^2} = \frac{2\alpha C^2}{(n-i)\sigma^2}\left(1 - \frac{2\eta\beta\rho}{\beta+\rho}\right)^{\frac{n-i+1}{2}} .
$$

This concludes the analysis of the case $i < n$.

For the case $i = n$ we need to bound $\mathsf{R}_\alpha(\xi K_0 \| \xi K_0')$, where now $\xi$ is the distribution of $x_{n-1}$, and the operators $K_0$ and $K_0'$ are defined as above. By Hölder's inequality, monotonicity of the logarithm, the contractiveness of $\Pi_\mathbb{K}$ and the Lipschitz assumption on $\ell$ we have

$$
\begin{aligned}
\mathsf{R}_\alpha(\xi K_0 \| \xi K_0') &\leq \sup_{x \in \mathrm{supp}(\xi)} \mathsf{R}_\alpha(K_0(x) \| K_0'(x)) \leq \sup_{x \in \mathbb{R}^d} \mathsf{R}_\alpha(K_0(x) \| K_0'(x)) \\
&\leq \sup_{x \in \mathbb{R}^d} \frac{\alpha\eta^2\|\nabla_x\ell(x, z_n) - \nabla_x\ell(x, z_n')\|^2}{2\eta^2\sigma^2} \leq \frac{2\alpha C^2}{\sigma^2} .
\end{aligned}
$$

$\square$

## C  Proofs for Section 5 (Diffusion Mechanisms)

**Theorem 6.** *Let $f : \mathbb{D}^n \to \mathbb{R}^d$ and let $\mathbf{P} = (P_t)_{t \geq 0}$ by a Markov semigroup on $\mathbb{R}^d$ satisfying Assumption 1. If the mechanism $M_t^f(D) = P_t(f(D))$ has intrinsic sensitivity $\Lambda(t)$, then it satisfies $(\alpha, \alpha\Lambda(t))$-RDP for any $\alpha > 1$ and $t > 0$.*

The proof of Theorem 6 relies, first of all, on the following lemma.

**Lemma 19.** *Let $\varphi : [t, \infty) \to \mathbb{R}$ be a function satisfying $\varphi(s) > 0$ and $\lim_{s \to \infty} \varphi(s) = 1$. Suppose there exists a function $\kappa(s)$ and a constant $c > 0$ such that for all $s \geq t$ we have $\frac{d}{ds}\varphi(s) \geq -c\kappa(s)\varphi(s)$. Then $\varphi(t) \leq \exp\left(c\int_t^\infty \kappa(s)ds\right)$.*

*Proof.* The bound follows from a direct application of the fundamental theorem of calculus. Indeed, noting $\lim_{s \to \infty} \log\varphi(s) = 0$, we have

$$
\begin{aligned}
-\log\varphi(t) &= \lim_{s \to \infty}\log\varphi(s) - \log\varphi(t) = \int_t^\infty \left(\frac{d}{ds}\log\varphi(s)\right)ds \\
&= \int_t^\infty \left(\frac{\frac{d}{ds}\varphi(s)}{\varphi(s)}\right)ds \geq -c\int_t^\infty \kappa(s)ds .
\end{aligned}
$$

$\square$

In order to apply this lemma to bound the Rényi DP of the diffusion mechanism $M_t^f$ we will need to compute the derivative with respect to $t$ of the Rényi divergence between $P_t(x)$ and $P_t(x')$. To be able to evaluate this derivative we will use some well-known relations between the kernel $p_t(x, y)$ of a semigroup with invariant measure $\lambda$ and its generator $L$, as well as further calculus rules for the carré du champ operator $\Gamma$. We now introduce the required properties without proof and recall they are standard facts in the theory of symmetric diffusion processes (see, e.g., [1]), and in particular they hold for any Markov semigroup satisfying Assumption 1.

1. (Reversible Fokker-Planck Equation) For any $x, y, t$ we have $\frac{d}{dt} p_t(x, y) = L_y p_t(x, y)$, where $L_y$ denotes the generator operating on $y \mapsto p_t(x, y)$.

2. (Integration by Parts) We have $\int \Gamma(f, g) d\lambda = - \int (Lf) g d\lambda$ for any $f, g$ where the integrals are defined.

3. (Chain Rule for $\Gamma$) For any differentiable function $\phi$ we have $\Gamma(\phi(f), g) = \phi'(f)\Gamma(f, g)$ for any functions $f, g$ where the terms are defined.

4. (Product Rule for $\Gamma$) We have $\Gamma(fg, h) = f\Gamma(g, h) + g\Gamma(f, h)$ for any functions $f, g, h$ where the terms are defined.

*Proof of Theorem 6.* Let us define the function $\phi(u) = u^\alpha$ for $\alpha > 1$ and note that the derivatives of $\phi$ satisfy the following identities:

$$\phi'(u) = \alpha \frac{\phi(u)}{u} \quad , \tag{7}$$

$$\phi''(u) = \alpha(\alpha - 1)\frac{\phi(u)}{u^2} \quad , \tag{8}$$

$$-u\phi''(u) = \frac{d}{du}\left(\phi(u) - u\phi'(u)\right) \quad . \tag{9}$$

Now fix datasets $D \simeq D'$ and let $x = f(D)$ and $x' = f(D')$. With this notation we have $M_t^f(D) = P_t(x)$, $M_t^f(D') = P_t(x')$ and $\mathsf{R}_\alpha(P_t(x) \| P_t(x')) = \frac{1}{\alpha - 1} \log \varphi(t)$, where we defined

$$\varphi(t) \triangleq \int \phi\left(\frac{p_t(x, y)}{p_t(x', y)}\right) p_t(x', y)\lambda(dy) \quad .$$

Since $\mathbf{P}$ has a unique invariant measure $\lambda$, then we must have $\lim_{t \to \infty} \frac{p_t(x,y)}{p_t(x',y)} = 1$ for any $x, y$, and therefore $\lim_{t \to \infty} \varphi(t) = 1$. Thus, by Lemma 19, to obtain the desired bound it suffices to show that the inequality $\frac{d}{dt}\varphi(t) \geq -\alpha(\alpha - 1)\kappa_{x,x'}(t)\varphi(t)$ holds for $t > 0$.

We will now show that this inequality is indeed satisfied. For simplicity, let use define the notation $p_t(y) \triangleq p_t(x, y)$, $q_t(y) \triangleq p_t(x', y)$, $r_t(y) \triangleq \frac{p_t(y)}{q_t(y)}$ and $\partial_t \triangleq \frac{d}{dt}$. With these, we now can apply the

properties of $\mathbf{P}$ and $\phi$ to compute the derivative of $\varphi(t)$ as follows:[10]

$$
\begin{aligned}
\partial_t \varphi(t) &= \int \partial_t \left( \phi(r_t) q_t \right) && \text{by Leibniz's rule ,} \\
&= \int \phi'(r_t)(\partial_t r_t) q_t + \phi(r_t)(\partial_t q_t) && \text{by calculus of } \partial_t \text{ ,} \\
&= \int \phi'(r_t) \frac{(L p_t) q_t - (L q_t) p_t}{q_t} + \phi(r_t)(L q_t) && \text{by Reversible Fokker-Planck Equation ,} \\
&= \int \phi'(r_t)(L p_t) + (\phi(r_t) - r_t \phi'(r_t))(L q_t) && \text{by re-arranging ,} \\
&= -\int \Gamma(\phi'(r_t), p_t) + \Gamma(\phi(r_t) - r_t \phi'(r_t), q_t) && \text{by Integration by Parts ,} \\
&= -\int \phi''(r_t)\Gamma(r_t, p_t) + \Gamma(\phi(r_t) - r_t \phi'(r_t), q_t) && \text{by Chain Rule for } \Gamma \text{ ,} \\
&= -\int \phi''(r_t)\Gamma(r_t, p_t) - r_t \phi''(r_t)\Gamma(r_t, q_t) && \text{by Chain Rule for } \Gamma \text{ and (9) ,} \\
&= -\alpha(\alpha-1) \int \frac{\phi(r_t)}{r_t^2}(\Gamma(r_t, p_t) - r_t \Gamma(r_t, q_t)) && \text{by (8) ,} \\
&= -\alpha(\alpha-1) \int \phi(r_t) q_t \left( \frac{q_t \Gamma(r_t, p_t) - p_t \Gamma(r_t, q_t)}{p_t^2} \right) && \text{by definition of } r_t \text{ .}
\end{aligned}
$$

The last step in the proof is to verify the following identify, which follows from the rules of calculus under $\Gamma$:

$$
\begin{aligned}
\Gamma(\log r_t, \log r_t) &= \frac{1}{r_t}\Gamma(r_t, \log r_t) && \text{by Chain Rule for } \Gamma \text{ ,} \\
&= \frac{1}{r_t^2}\Gamma(r_t, r_t) && \text{by Chain Rule for } \Gamma \text{ ,} \\
&= \frac{1}{r_t^2}\Gamma\left( r_t, \frac{p_t}{q_t} \right) && \text{by definition of } r_t \text{ ,} \\
&= \frac{1}{r_t^2}\left( \frac{1}{q_t}\Gamma(r_t, p_t) + p_t \Gamma\left( r_t, \frac{1}{q_t} \right) \right) && \text{by Product Rule for } \Gamma \text{ ,} \\
&= \frac{1}{r_t^2}\left( \frac{1}{q_t}\Gamma(r_t, p_t) - \frac{p_t}{q_t^2}\Gamma(r_t, q_t) \right) && \text{by Chain Rule for } \Gamma \text{ ,} \\
&= \frac{q_t \Gamma(r_t, p_t) - p_t \Gamma(r_t, q_t)}{p_t^2} && \text{by definition of } r_t \text{ .}
\end{aligned}
$$

Now we finally put the last two derivations together to conclude that

$$
\begin{aligned}
\frac{d}{dt}\varphi(t) &= -\alpha(\alpha-1) \int \phi\left( \frac{p_t(x,y)}{p_t(x',y)} \right) p_t(x',y) \Gamma\left( \log \frac{p_t(x,y)}{p_t(x',y)} \right) \lambda(dy) \\
&\geq -\alpha(\alpha-1)\kappa_{x,x'}(t) \int \phi\left( \frac{p_t(x,y)}{p_t(x',y)} \right) p_t(x',y)\lambda(dy) \\
&= -\alpha(\alpha-1)\kappa_{x,x'}(t)\varphi(t) \ .
\end{aligned}
$$

$\square$

**Corollary 7.** *Let* $f : \mathbb{D}^n \to \mathbb{R}^d$ *have global* $L_2$-*sensitivity* $\Delta$ *and* $\mathbf{P} = (P_t)_{t\geq 0}$ *be the Ornstein-Uhlenbeck semigroup with parameters* $\theta, \rho$. *For any* $\alpha > 1$ *and* $t > 0$ *the mechanism* $M_t^f(D) = P_t(f(D))$ *satisfies* $(\alpha, \alpha\Lambda(t))$-*RDP with* $\Lambda(t) = \frac{\theta\Delta^2}{2\rho^2(e^{2\theta t}-1)}$.

*Proof.* Using the expression of the kernel of $P_t$ with respect to the invariant measure $\lambda$ we first compute

$$\log\left(\frac{p_t(x, y)}{p_t(x', y)}\right) = \frac{\theta e^{\theta t}\langle x - x', y\rangle}{\rho^2(e^{2\theta t} - 1)} \quad .$$

Next we use the expression $\Gamma(f) = \rho^2\|\nabla f\|^2$ for the carré du champ operator to obtain

$$\kappa_{x,x'}(t) = \frac{\theta^2 e^{2\theta t}\|x - x'\|^2}{\rho^2(e^{2\theta t} - 1)^2} \quad .$$

Applying the easily verifiable integral formula

$$\int_t^\infty \frac{e^{2\theta s}}{(e^{2\theta s} - 1)^2}ds = \frac{1}{2\theta(e^{2\theta t} - 1)}$$

in the definition of $\Lambda(t)$ yields the desired result. $\qquad\square$

**Theorem 8.** *Suppose $f : \mathbb{D}^n \to \mathbb{R}^d$ has global $L_2$-sensitivity $\Delta$ and satisfies $\sup_D \|f(D)\| \le R$. If $\theta R^2 \le 4d\rho^2$ then we have $\frac{\mathcal{E}_{\mathrm{OU}}(\theta,\rho,t)}{\mathcal{E}_{\mathrm{GM}}(\theta,\rho,t)} \le 1$ for all $t \ge 0$ and $\lim_{t\to\infty} \frac{\mathcal{E}_{\mathrm{OU}}(\theta,\rho,t)}{\mathcal{E}_{\mathrm{GM}}(\theta,\rho,t)} = 0$. In particular, taking $\theta = \log\left(1 + \frac{d\Delta^2}{2\epsilon R^2}\right)$ and $\rho^2 = \frac{\theta\Delta^2}{2\epsilon(e^{2\theta}-1)}$ with $\epsilon > 0$, the mechanism $M_t^f$ satisfies $(\alpha, \alpha\epsilon)$-RDP at time $t = 1$ and we have $\frac{\mathcal{E}_{\mathrm{OU}}(\theta,\rho,1)}{\mathcal{E}_{\mathrm{GM}}(\theta,\rho,1)} \le \left(1 + \frac{d\Delta^2}{2\epsilon R^2}\right)^{-1}$.*

*Proof.* First note that at time $t = 0$ we have $\mathcal{E}_{\mathrm{OU}}(\theta, \rho, 0) = \mathcal{E}_{\mathrm{GM}}(\theta, \rho, 0) = 0$. Thus, to see that $\mathcal{E}_{\mathrm{OU}}(\theta, \rho, t) \le \mathcal{E}_{\mathrm{GM}}(\theta, \rho, t)$ for $t > 0$ it is enough to check that $\frac{d}{dt}\mathcal{E}_{\mathrm{OU}}(\theta, \rho, t) \le \frac{d}{dt}\mathcal{E}_{\mathrm{GM}}(\theta, \rho, t)$ for $t \ge 0$. Indeed, differentiating (4), this follows from the boundedness of $f$ and $\theta R^2 \le 4d\rho^2$ by noting:

$$\frac{d}{dt}\mathcal{E}_{\mathrm{OU}}(\theta, \rho, t) \le 2d\rho^2 e^{-2\theta t} + 2\theta R^2 e^{-\theta t}(1 - e^{-\theta t}) \le 2d\rho^2 e^{-2\theta t} + 8d\rho^2 e^{-\theta t}(1 - e^{-\theta t})$$

$$= 2d\rho^2 e^{-2\theta t}(4e^{\theta t} - 3) \le 2d\rho^2 e^{2\theta t} = \frac{d}{dt}\mathcal{E}_{\mathrm{GM}}(\theta, \rho, t) \quad ,$$

where the last two steps use the inequality $4e^s - 3 \le e^{4s}$, $s \ge 0$, and the definition of $\tilde{\sigma}^2$. To see that the ratio converges to 0 we just observe that the limit of $\mathcal{E}_{\mathrm{OU}}(\theta, \rho, t)$ is finite while $\mathcal{E}_{\mathrm{GM}}(\theta, \rho, t)$ grows to infinity as $t \to \infty$.

The privacy bound in the case with a fixed level of privacy at $t = 1$ follows from directly from Corollary 7. The error bound follows substituting the chosen parameters in the expression for the mean squared error. In the first place, we use the definitions of $\tilde{\sigma}^2$ and $\rho^2$ to get

$$\mathcal{E}_{\mathrm{GM}}(\theta, \rho, 1) = d\tilde{\sigma}^2 = \frac{d\rho^2(e^{2\theta} - 1)}{\theta} = \frac{d\Delta^2}{2\epsilon} \quad .$$

On the other hand, substituting the choice for $\rho$ on the error of the Ornstein-Uhlenbeck mechanism and using the boundedness of $f$ we get

$$\mathcal{E}_{\mathrm{OU}}(\theta, \rho, 1) \le (1 - e^{-\theta})^2 R^2 + \frac{d\rho^2}{\theta}(1 - e^{-2\theta}) = (1 - e^{-\theta})^2 R^2 + \frac{d\Delta^2}{2\epsilon}e^{-2\theta} \quad .$$

Finally, plugging the choice of $\theta$ in this last expression yields:

$$(1 - e^{-\theta})^2 R^2 + \frac{d\Delta^2}{2\epsilon}e^{-2\theta} = \frac{R^2\left(\frac{d\Delta^2}{2\epsilon R^2}\right)^2 + R^4\frac{d\Delta^2}{2\epsilon R^2}}{\left(R^2 + \frac{d\Delta^2}{2\epsilon}\right)^2} = \frac{d\Delta^2}{2\epsilon}\frac{1}{1 + \frac{d\Delta^2}{2\epsilon R^2}} \quad .$$

$\qquad\square$

## Footnotes

[9] Here $\Pi_{\mathbb{K}}(x) = \arg\min_{y\in\mathbb{K}} \|x - y\|$ denotes the projection operator onto the convex set $\mathbb{K} \subseteq \mathbb{R}^d$.

[10] All integrals in this derivation are with respect to the invariant measure $d\lambda$, which is omitted for convenience.