[Reviews · NeurIPS 2019]

Reviewer 1



This work shows how a number of different Markov operators can amplify of the privacy of a DP mechanism when used in post-processing. First, results about amplification under several different uniform mixing conditions (of varying strengths) are provided. Next, results about Renyi DP (from which can be obtained results about (eps,delta)-DP) that take into account couplings between measures on the data space are provided; the most basic result yields a decomposition that makes data-processing inequality evident. This result is leveraged iteratively (inductively) to show that projected SGD + noise on a strongly-convex objective function, run for O(log(d)) more iterations, has the same optimization error as non-private SGD while ensuring privacy for all records. Finally, post-processing mechanisms based on diffusions are examined; a Renyi-DP guarantee for operators having a symmetric kernel w.r.t. their invariant measure is provided. Examples are provided throughout, including for Brownian motion vs. Ornstein-Uhlenbeck mechanisms, with superior DP guarantees of the latter established. The paper is well written, and the numerous examples are both helpful for general understanding and illustrate the relevance of the results. Theorem 4 (on DP of noisy SGD) is a nice result with a clean proof, but “noisy SGD” is not the optimal algorithm to consider (as SGD already has noise from the gradients); the use of this algorithm is a limitation of the analysis, and it would be nice for it to be eventually removed. Also, for this result, it may be useful to state instead that it holds for maps \psi_i’s that are contractions, rather than for general Lipschitz \psi_i’s; the bound is vacuous otherwise. [UPDATE] Upon closer review of the literature, it appears that the paper greatly leverages the techniques and results of Feldman et al. (2018). E.g., the purported exponential improvement over their bound in Thm 5 (Thm 23, resp.) in specifically the strongly convex case uses a standard result from convex optimization, which could as well have been used in their (somewhat more streamlined) analysis had they chosen to separately handle the strongly convex case. The paper does a great job of synthesizing and presenting DP ideas for readers who may be more familiar with diffusion processes than DP, but it is written in a way that suggests that the techniques (rather than mostly the perspective) are new; this is disappointing. Novel DP applications would go some way towards making up for that.

Reviewer 2



The paper deepens our understanding of the relationship between differential privacy and theory of stochastic processes. The main reason for exploring the connection is refined analyses (also known as "amplification theorems") of iterated mechanisms, which is of great relevance to the ML community. If early differentially private mechanisms targeted one-shot queries, learning problems call for iterative versions of DP mechanisms. The paper builds upon a recent work by Feldman et al. on privacy amplification-by-iteration where the main technical tool was a statement between closeness of distributions over Banach spaces. The submission generalizes this result using the measure-theoretic language of Markov operators and their properties. While, by itself, this new approach does not not change the underlying mechanism, it does lead to stronger guarantees in the case of strong convexity, and paves the way for more interaction between these areas of research.

Reviewer 3



I think this is a fascinating paper. I really like the idea of finding a unifying framework for analyzing the privacy-loss after post-processing, and I think that the fairly long list of running examples throughout the paper suggests how applicable this framework is. The view of post-processing (randomly) as a Markov process is novel (to me at least), the paper is fairly clear (the parts that I found puzzling were mostly regrading background material which I do not know), and I deem this paper to be very significant. I thus advocate acceptance. Minor comments: (1) It would have been helpful had your first example been subsampling, in Section 3, clarifying precisely which of the 4 definitions subsampling satisfies. (#4) (2) Line 139: by \lambda you me \nu, right? (Otherwise the coupling \pi\in{\cal C}(\mu,\nu) is defined in a way independent of \nu...) (3) Line 228: Ito?!? (4) All throughout I thought of K as a computation --- taking x (the output of a DP mechanism) and some \omega (a random string) and producing y. I am still unclear as to whether this interpretation misses something... (5) Section 5, perhaps because it is replete with terminology foreign to me, was hard to follow. In particular, I do not understand WHY / WHEN should one view certain post-processing algorithms as these chain of mechanisms, nor is it clear to me WHY/WHEN would someone choose to use the mechanism of Section 5.1. A motivating example could be useful, if you can think of one. Afterall -- the way *I* view it (and I could be wrong) --- the goal is not to think of DP-mechanisms as mechanisms for mathematical sake, but rather as DP-mechanisms that ought to be useful for some computational tasks. ----------------------------------------------------------------------------------- Post rebuttal: there have been serious concerns raised by the Senior PC member regarding the applications of the proposed framework. As a result, I too am reducing slightly my score. However, I still am very much fond of the overall approach that unifies several techniques under one framework and see significant merit to this paper. Thus, I still advocate acceptance, and I genuinely believe that NeurIPS should have this paper in its proceedings.

Reviewer 4



The main motivation of this work is to provide a general treatment of the privacy amplification properties of post-processing mechanisms. Three unrelated types of such postprocessing mechanisms are considered. 1. One step mixing. Translate closeness properties for postprocessing of individual inputs into amplification. Unfortunately, the authors do not give any examples of how these results could lead to a better analysis. On a technical level these results are fairly standard calculations. 2. The coupling technique. They give a somewhat more general formulation of the technique from [Feldman et al 2018]. It is then shown that the technique can be applied with Laplace noise and also gives better bounds in the smooth and strongly convex case. In this case the iteration is strictly contractive. As far as I can tell both of these applications do not require the more general statement and are straightforward applications of the technique from [Feldman et al, 2018]. The results are stated in the abstract and the intro in a way that suggests that these are applications of the techniques introduced in this work. I find this misleading. 3. Diffusion-based mechanisms. I found the presentation of this section to be unfriendly to those not familiar the relevant concepts from statistical physics. In particular, I could not find a way to convert the description given in this work into actual algorithms. I think it would need significant rewriting to be readable by most privacy researchers. As applications they rederive Gaussian mechanism and another mechanism (referred to as Ornstein-Uhlenbeck) that is not described explicitly. The verbal description of its properties are easy to achieve by postprocessing the Gaussian mechanism. Slight shrinkage is known to reduce l_2 error in high dimensions (Stein's phenomenon) and over bounded domain it is safe to do l_2 projection to the domain (which ensures boundedness of the error). Overall while the work has some nice observations and some of the formalism might end up being useful, the work does not provide any evidence that their results might lead to useful and new privacy analyses.

[Author Response · NeurIPS 2019]

1   We thank all reviewers for their praise and constructive criticism.

2   We will revise the manuscript to address the presentation issues flagged up in the reviews. In particular, we will:

3       • Emphasize that Theorem 4 is mostly useful when the maps $\psi_i$ are contractions.

4       • Include further background material on the *carre du champ* operator, probably in the appendix.

5       • Provide a complementary presentation of the Ornstein-Uhlenbeck mechanism aimed at practitioners.

6       • Include examples of mechanisms satisfying the conditions of Section 3.

7       • Expand the details about the construction of the transport Markov operator.

8       • We will try to find a compelling application for the mechanisms presented in Section 5.

[Meta-Review · NeurIPS 2019]

The work gives a collection of results on privacy amplification by postprocessing. The results reveal some interesting connections. Further, the work gives a refinement of the analysis from Feldman et al, 2018 work on privacy amplification by iteration in the strongly convex case. While the initial reviews were very strong, an additional solicited review by an expert has brought up significant issues. Specifically, misrepresentation of the strongly convex case as a new application, lack of any other new applications and poor presentation in the section on diffusion. While the eventual decision is to accept the work these issues should be addressed as well as possible in the final version.